# Delegated Classification

**Eden Saig,    Inbal Talgam-Cohen,    Nir Rosenfeld**
Technion – Israel Institute of Technology
Haifa, Israel
{edens,italgam,nirr}@cs.technion.ac.il

## Abstract

When machine learning is outsourced to a rational agent, conflicts of interest might arise and severely impact predictive performance. In this work, we propose a theoretical framework for incentive-aware delegation of machine learning tasks. We model delegation as a principal-agent game, in which accurate learning can be incentivized by the principal using performance-based contracts. Adapting the economic theory of contract design to this setting, we define *budget-optimal* contracts and prove they take a simple threshold form under reasonable assumptions. In the binary-action case, the optimality of such contracts is shown to be equivalent to the classic Neyman-Pearson lemma, establishing a formal connection between contract design and statistical hypothesis testing. Empirically, we demonstrate that budget-optimal contracts can be constructed using small-scale data, leveraging recent advances in the study of learning curves and scaling laws. Performance and economic outcomes are evaluated using synthetic and real-world classification tasks.

## 1   Introduction

The acclaimed success of machine learning at effectively solving difficult prediction tasks across diverse problem domains has made it highly appealing for firms, institutions, and individual practitioners. But machine learning has also become increasingly complex, cumbersome, and difficult to operate—and not all those who seek to learn have access to the necessary expertise, infrastructure, and designated resources required for learning effectively. This gap has created a new market for *outsourced machine learning*, in which a client interested in obtaining an accurate predictive model can hire the services of a specialized provider which, for a price, trains the model on their behalf. Consider for example a hospital purchasing a classifier for deciding between hospitalization and outpatient treatment when triaging patients. The provider invests in curating, cleaning and annotating training data, and delivers a trained model in return to payment from the hospital.

Having a budget to expend on outsourced learning [51], we model the client as aiming to obtain the best possible predictive model. At first glance, it is tempting to assume that the optimal strategy is simply to pay the provider the maximal feasible amount—and hope to get a high-end model in return. After all, if the client were to spend the budget directly on learning, investing the maximal available sum would yield the best possible results. But this neglects to account for the *incentives* of the provider, who is interested in maximizing profit. Since the actions of the provider remain private, it is in his best interest to (secretly) minimize efforts, which in turn can result in his delivering a suboptimally-trained model. In our example, the provider can cut costs by annotating only a subset of the data, obtaining cheaper low-quality annotations, or neglecting to meticulously remove all outliers.

Outsourced learning is hence susceptible to *moral hazard*, an economic situation which might occur under information asymmetry, and to the detriment of the client. Motivated by this observation, in this paper we initiate the study of *delegated learning*, and aim to explore the economic, algorithmic, and statistical implications that occur when learning is delegated to a specialized provider. Our key novelty is in instantiating delegated learning as a problem of *optimal contract design* [10, 42, 47, 48]. Broadly, contracts are an important monetary device that allows the client to establish a payment scheme

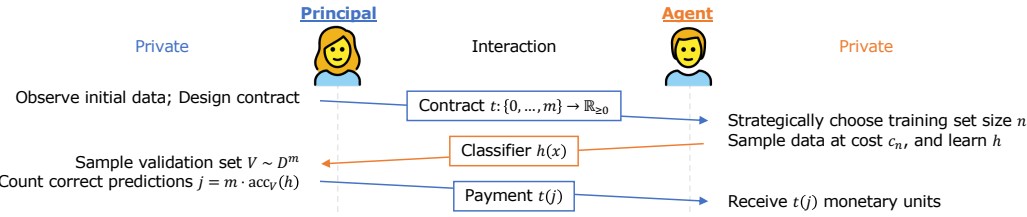

Figure 1: Delegated classification interaction sequence. The principal examines initial information, and designs a contract $t : \{0, \dots, m\} \to \mathbb{R}_{\geq 0}$. Having observed $t$, the agent strategically selects a dataset size $n$ that will maximize his expected utility. He samples a training set $S \sim D^n$, incurs cost $c_n$, then trains the classifier $h \in \mathcal{H}$ and sends it to the principal. Upon receiving $h$, the principal evaluates its accuracy on a random validation set $V \sim D^m$, and pays the agent according to the contract $t$.

which, if properly set, serves to align incentives and guarantee that both parties are well-off. Our main challenge is to design effective contracts specialized to the task of delegated learning on a budget.

Towards this, we begin with a conventional supervised classification setup, and impose economic structure by assuming that acquiring training examples is costly. We then conceptually "split" the conventional self-sufficient learner into two rational entities: a *principal*, who controls the budget and is interested in maximizing predictive accuracy; and an *agent*, who controls learning (in particular the training set) and is interested in maximizing profit. This allows us to model principal-agent relations as a Stackelberg game, in which the principal commits to a *contract* $t$, determining *a priori* the amount to be paid for every possible (stochastic) level of obtained accuracy. The agent best-responds to the contract by choosing the profit-maximizing number of samples $n$, and training the predictive model.

Under this setting, we study the algorithmic problem of designing an optimal contract. As is standard in economic analysis, we begin with the assumption that the principal has full information on the distribution of possible outcomes for each of the agent's possible actions. In our setting, actions correspond to the number of training samples, and outcomes to the empirical classifier accuracy; thus, the main object of interest for contract design in delegated learning settings is the *learning curve*, which describes the (stochastic) performance of learning per sample size. Under certain plausible conditions on the learning curve, namely MLRP and a certain notion of concavity, our main result here is that optimal contracts are *simple*, and in particular, take on the form of simple threshold functions. Simple contracts are appealing because they are straightforward to understand and communicate; in our setting, they are also easy to compute, and we give a closed-form solution for the optimal threshold contract. Providing an interpretation of the closed-form solution, our findings establish a new connection between contracts and the renowned Neymon-Pearson lemma [45], which we consider as one of our central theoretical contributions.

We then switch gears and turn to empirically studying the construction of contracts from partial information. In particular, we consider a setting where the principal can only estimate the learning curve from small available data (e.g., by bootstrapping on small $n$ and extrapolating). Using the recent LCDB dataset of learning curves [43], we show that threshold contracts generally perform well on estimated curves despite the inherent uncertainty. We also explore the role of different parameters of our setup, consider various tradeoffs in curve-fitting and contract design, and discuss limitations by pointing out certain failure modes to which contracts may be susceptible.

Taken together, our results shed light on why and how simple contracts for delegated learning work to correctly balance between the incentives of both delegator and delegatee in outsourced learning.

## 1.1 Related work

Previous works have considered delegation of ML-related tasks that differ from our task of training a classifier: labeling of data points in [14], gathering information in [16], and computing a costly high-dimensional function in [5]. In the delegated task of [24], the agent provides a classifier and the principal verifies its near-optimality within $\mathcal{H}$. A fundamental difference is that their agent is assumed to be adversarial rather than rational, and so interactive proofs are used instead of economic incentives.

The Neyman-Pearson lemma has recently been connected to economic design by [8] in the context of adverse selection rather than moral hazard. The agent has a hidden type (e.g., whether a new

drug is effective), and the optimal menu to offer this agent is designed based on the theory of e-values. When the hidden type has binary support, the method is equivalent to Neyman-Pearson. A "moral" link between the design of contracts (for a non-budgeted principal) and statistical inference (in particular likelihood ratios) was observed already in [25], but no connection was made to the power of hypothesis tests. Other intersections of ML and contracts that do not involve delegation of learning-related tasks include strategic classification [e.g., 36, 37, 3] and online learning of optimal contracts [29, 17, 52] Contract settings with a binary action and/or outcome space have been studied in [e.g., 20, 7, 22]. Not to be confused with our notion of delegation, there is a growing computational literature on delegation without monetary payments [e.g., 35].

To extrapolate from partial data, our work builds upon recent advancements in the study of learning curves, which characterize the expected generalization of learning as a function of dataset size and other exogenous factors [49]. There is growing empirical evidence that performance of modern neural networks can be predicted using simple scaling laws [e.g., 34, 41, 50, 23, 2, 46, 30], and theoretical results that back these findings in simplified settings [11, 12, 33, 9].

Perhaps closest to ours is the concurrent work [4], which analyzes a similar setting under different assumptions: Rather than assuming budget constraints, they assume that the principal's utility is linear in accuracy and payout, and the learning curve takes a specific functional form. Based on these assumptions, they derive approximately optimal linear contracts which are robust to adverse selection. In contrast, we obtain globally optimal simple contracts based on hypothesis testing, and present data-driven methods to handle partial information. Together, the two studies demonstrate the important role of simple contracts in the growing ecosystem of machine learning delegation.

## 2 Problem Setup

The core of our setting is based on a standard supervised classification task. Let $x \in \mathcal{X}$ be features and $y \in \mathcal{Y}$ be labels, and assume there is some unknown joint distribution $D$ over $(x, y)$ pairs. Given a sample set $S = \{(x_i, y_i)\}_{i=1}^n \sim D^n$, the goal in learning is to use $S$ to find a classifier $h : \mathcal{X} \to \mathcal{Y}$ from a class $\mathcal{H}$ that maximizes expected accuracy, $\mathrm{acc}_D(h) = \mathbb{P}_{(x,y) \sim D}[h(x) = y]$. Because the underlying distribution $D$ is unknown, expected performance is estimated by the empirical average on an additional held-out validation set $V \sim D^m$ of size $m$, as $\mathrm{acc}_V(h) = \frac{1}{m} \sum_{i=1}^m \mathbb{1}[h(x_i) = y_i]$, which is a consistent and unbiased estimator of $\mathrm{acc}_D(h)$. We will assume throughout that both the learning algorithm and the validation set size $m$ are known and fixed.

**Learning on a budget.** We will be interested in studying learning when certain resources are limited or costly. Our main focus will be on the setting where the main cost of learning is the number of labeled examples $n$, but we note that our approach can in principle extend to other forms of learning 'effort'.[1] We assume the learner has a monetary budget $B$ to spend on samples, and is interested in maximizing accuracy under budget constraints. Let $c_n \geq 0$ be the cost of $n$ samples (assumed to be increasing in $n$), then the learner aims to solve:

$$n^* = \mathrm{argmax}_n \, \mathbb{E}_{h_n}[\mathrm{acc}_D(h_n)] \quad \text{s.t.} \quad c_n \leq B \tag{1}$$

where $h_n$ is a classifier learned from a random dataset of size $|S| = n$. Note that $h_n$ is a random variable with distribution depending on $n$. We denote the out-of-sample accuracy of $h_n$ by $\alpha_n = \mathrm{acc}_D(h_n)$. When the learner is a self-sufficient entity, and when the expected $\alpha_n$ improves monotonically in $n$, then $n^*$ in Eq. (1) in naturally the largest affordable $n$ (see Sec. 3). However, as we will see, when learning is *delegated*—this seemingly straightforward observation can break.

**Delegation.** We model the delegation of learning as a conceptual partition of the learner into two distinct entities: an *agent*, who controls learning; and a *principal*, who controls the validation process. The principal outsources the learning task to the agent, who in turn uses the training set $S$ to train the classifier $h$; once delivered, the principal validates the performance of $h$ using the validation set $V$. Whereas the classifier's accuracy benefits the principal alone, the cost of learning (i.e., the cost of acquiring $S$) is born exclusively by the agent. Importantly, the amount of invested effort remains private to the agent; in our example, the principal cannot know how many examples received quality labeling. Because the agent seeks to maximize profit, the principal can use her budget as a source of monetary payment to incentivize the agent to invest in larger $|S| = n$. Intuitively, one could expect

---

[1]For example, [34] argue that not only training-set size, but also computation time and model size are related to accuracy through scaling laws, which have tight connections to the assumptions we discuss in Sec. 3.

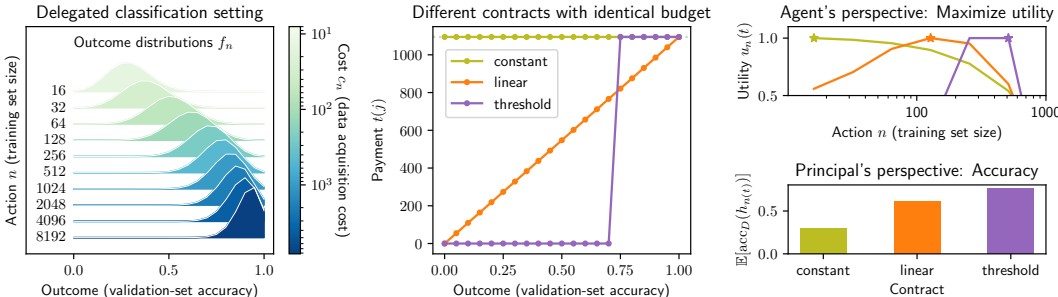

Figure 2: A delegated classification setting (data from Sec. 4). **(Left)** Each costly action taken by the agent (training set size $n$) induces a distribution $f_n$ of possible outcomes (classifier accuracy). The principal seeks to construct a contract $t$ that incentivizes a profit-maximizing agent to take actions entailing favorable outcomes. Note the $f_n$ exhibit increasing expectation, but decreasing variance, in $n$. **(Center)** Three contracts for a given budget $B$, mapping outcomes to payments. **(Top-right)** Agent's utilities $u_n(t)$ and best responses $n(t)$ (stars) for each contract $t$. **(Bottom-right)** Expected accuracies for principal resulting from each contract; here the threshold contract is optimal (see Sec. 3).

larger payments to entail larger $n$, and therefore higher-accuracy $h$. However, as we will see, this is not always the case, and careful planning is required in order to fully utilize a given budget.

## 2.1 Delegation as contract design

As the training set remains private to the agent, there is an information gap between the two parties. This creates a conflict of interest for the agent known as *moral hazard* [10], in which the agent may be tempted to invest sub-par effort, while claiming that efforts were in fact his honest best. In economics, the celebrated solution to moral hazard are *contracts* [31]: pay-per-performance rules that a-priori determine future payments for every possible outcome, which we formally describe next.

**Contract design.** A contract setting is defined by a set of actions $\mathcal{A} = \{a_1, \ldots, a_N\}$ that can be taken by the agent, and a set of possible outcomes $j \in \{0, \ldots, m\}$. Each action $a_i$ is associated with a cost $c_i$, and w.l.o.g. we assume $c_1 \leq \cdots \leq c_N$ so that actions correspond to increasing *effort levels*. The agent's choice to perform action $a \in \mathcal{A}$ yields a random outcome $j \sim f_a$ for the principal, where $f_a$ describes a distribution over the possible outcomes associated with action $a$. The principal, who does not observe the agent's chosen action, can incentivize the agent through a *contract*, $t : \{0, \ldots, m\} \to \mathbb{R}_{\geq 0}$, according to which she pays the agent $t(j) \geq 0$ when the materialized outcome is $j$. Given contract $t$, let $u_a(t)$ be the agent's expected *utility* from taking action $a \in \mathcal{A}$ at cost $c_a$ (via stochastic outcomes $j \sim f_a$), and let $a(t)$ be the agent's *best response*—an action that maximizes his expected utility (and following standard tie-breaking assumptions as in [18]). Then:

$$u_a(t) = \mathbb{E}_{j \sim f_a}[t(j)] - c_a, \qquad a(t) \in \operatorname{argmax}_{a \in \mathcal{A}} u_a(t). \qquad (2)$$

Every action $a^*$ that is the best response $a^* = a(t)$ to some contract $t$ is called *implementable*. In economic terms, the principal and agent are playing a *Stackelberg game*, in which the principal commits to a contract $t$ and the agent best-responds by choosing action $a(t)$ that maximizes his expected utility $u_a(t)$. The goal of the principal is to design a contract $t$ which incentivizes the agent to take best-response actions yielding favorable outcomes for the principal.

**Contracts for delegated learning.** We propose to formulate delegated learning as a problem of optimal contract design, instantiated as follows. First, we relate agent actions $a$ with the number of samples $n$, and denote $\mathcal{A} = \{n_1, \ldots, n_N\}$ as the possible sizes of $S$ that the learning agent can work with. The cost of acquiring samples naturally maps as $c_a = c_n$, and agent's best response is $a(t) = n(t)$. Next, we associate outcomes $j$ with accuracy for the principal by defining $j$ as the number of validation samples (out of the possible $m$) on which $h$ is correct; note this implies $\operatorname{acc}_V(h) = j/m$, and we will therefore use $j$ and $\operatorname{acc}_V(h)$ as 'outcomes' interchangeably. Finally, for an action $n$, we set $f_n$ to be the distribution over possible accuracies obtained when learning with $n$ samples, namely $f_n(j) = \mathbb{P}_{h_n, V}[\operatorname{acc}_V(h_n) = j/m] \; \forall j$. We will also use the matrix form $F_{nj} = f_n(j)$, where $F \in [0, 1]^{N \times (m+1)}$. Note that $F$ admits two sources of variation: (i) *a-priori* variation in $h_n$ due to stochasticity in $S \sim D^n$; and (ii) *a-posteriori* variation in $j$ for any fixed $h_n$

due to stochasticity in $V \sim D^m$. When $h_n$ is fixed, the outcome distribution admits a simple binomial form, namely $j \sim \text{Binomial}(m, \alpha_n)$. Empirically, we observe this to be the dominant component.

## 2.2 Delegation as an optimization problem

**Budget-optimal contracts.** Recall that the principal seeks to maximize accuracy under budget constraints (Eq. (1)). Once learning is delegated to an agent and framed as a contract design problem, the principal's objective becomes:

$$t^* = \text{argmax}_{t \in [0,B]^m} \mathbb{E}_{h_{n(t)}}[\text{acc}_D(h_{n(t)})] \tag{3}$$

Contract $t^*$ is chosen to incentivize the agent to invest effort $n(t)$ (via Eq. (2)) such that the training of $h_{n(t)}$ yields high dividends for the principal in terms of expected accuracy. We will refer to $t^*$ as a *budget-optimal contract*, and to the general task of finding $t^*$ as *budget-optimal contract design*.

**Information structure.** Delegated learning settings have actions $n$ and costs $c_n$ known to both sides, and outcome distribution $F_{nj}$ known to the agent. For the principal, we explore varying levels of knowledge: In Sec. 3, we assume (as in the classic contract design literature) that the principal has full information of $F$ (i.e., knows the learning curve), and focus on characterizing the optimal contract. In Sec. 4 we relax this assumption, and explore a partial-information setting in which the principal relies instead on an empirically-estimated curves $\hat{F}$.

# 3 Budget-Optimal Contract Design

## 3.1 The problem

**Why agents cut corners.** The conceptual challenge in designing contracts lies in that agents cannot reliably report *what* they did. For example, consider a principal who, after delegation, received a classifier attaining 0.74 (validation) accuracy. Should she be happy? The crux is that there are two ways that this could have happened: (i) the agent invested high effort in learning (large $n$), but received an uninformative $S$ by chance, and delivered a low-quality $h$ as a result; and (ii) the agent invested low effort (small $n$). Since the agent's actions are private, and because outcomes are stochastic, the principal can never know for certain which is the true underlying cause. In other words, a 'lazy' (or rather strategic) agent can hide behind the uncertainty that is inherent in learning outcomes.[2]

**Contract types.** To overcome this informational gap, the principal must devise a contract to align incentives and encourage the agent to prefer certain actions over others. But not all contracts are equally effective. Fig. 2 illustrates for a budget $B$ three contract types and their economic implications:

- **Constant contract** ($t(j) = B$): The agent is paid $B$ regardless of the outcome. His best-response in this case is to choose the least-costly action—to the detriment of the principal.
- **Linear contract** ($t(j) = Bj/m$): The agent is paid a fraction of $B$, linear in the resulting accuracy. Linear contracts are a popular and extensively-studied class of contracts [e.g., 32, 15]. Nonetheless, and though seemingly sensible, linear contracts turn out to be sub-optimal for our setting.
- **Threshold contract** ($t(j) = B\mathbb{1}[j \geq j_0]$ for some $j_0$): The agent is paid $B$ provided the empirical accuracy surpasses a threshold $j_0$. In the example in Fig. 2, the threshold contract is optimal.

Rather than committing *a-priori* to some type of contract, we seek to find the best budget-optimal contract by solving Eq. (3). For this it is useful to have *structure*.

**Stochastic learning curves (and where to find them).** Our approach uses the observation that there is a tight connection between the set of distributions $\{f_n\}$ encoded in $F$, and *learning curves*, which describe the anticipated accuracy of a classifier as a function of the size of its training set. Learning curves typically depict only expected accuracy, but there is also inherent variation in outcomes. We will therefore broadly use the term 'stochastic learning curve' to describe both mean trend *and* variation in accuracy as a function of $n$; formally, a stochastic learning curve is defined precisely by $F$. This connection is useful because learning curves have structure: First, expected learning curves are typically *monotone* [34, 11]; when not [43, 49], they can be monotonized [12]. Second, stochastic learning curves are likely to satisfy the *monotone likelihood ratio property* (MLRP), which states that the better the performance of a classifier, the more likely it was trained on more data (see Def. 1).

---

[2]The agent can also hide in the uncertainty due to $V$, particularly when $m$ is small; see experiment in Sec. 4.2.

Table 1: Characterization of simple min-budget contracts in different settings. Simple contract forms include *all-or-nothing contracts* and their subclass of *threshold contracts*. The table specifies for each configuration either the simple form that is optimal (in one case through equivalence to the Neyman-Pearson lemma), or that the simple form is non-optimal or intractable.

| Problem size | | Structural assumptions | | |
|---|---|---|---|---|
| Actions | Outcomes | No assumptions | MLRP | Concave-MLRP |
| $\|\mathcal{A}\| = 2$ | any size | All-or-nothing (T1) | Threshold (B.7.1) | |
| $\|\mathcal{A}\| > 2$ | $m + 1 = 2$ | All-or-nothing (B.5.1) | Threshold (B.7.2) | |
| | $m + 1 > 2$ | Simple is NP-hard (T3) | $\exists$ non-threshold (B.7.3) | Threshold (T4) |

## 3.2 Optimization via min-budget contracts

Our main technique for solving Eq. (3) relies on a reduction to what we refer to as *min-budget contracts*. Given an (implementable) target action $n^* \in \mathcal{A}$, a min-budget contract for $n^*$ is a contract $t$ that incentivizes the agent to employ precisely the action $n^*$, while minimizing the maximum payment by the principal $\|t\|_\infty = \max_{j \in \{0,\dots,m\}}\{t(j)\}$; i.e., $t$ implements $n^*$ at minimum budget. Formally:

$$t^* = \operatorname{argmin}_t \|t\|_\infty \quad \text{s.t.} \quad n(t) = n^* \tag{4}$$

Our reduction relies on the following claim (Proof in Appendix B.1):

**Proposition 1.** *Every budget-optimal contract design problem has an optimal solution which is also min-budget.*

Using Prop. 1, a solution to Eq. (3) can be obtained by iteratively solving Eq. (4): For all $n_i \in \mathcal{A}$, solve Eq. (4) with target action $n^* = n_i$, and return $t^*(n_i)$ for the best implementable $n_i$ whose budget does not exceed $B$. The budget-optimal problem thus reduces to solving multiple min-budget problems.

To compute each $t^*(n_i)$ in Eq. (4), we formulate a novel MIN-BUDGET linear program (LP),[3] detailed in Appx. B.2. One way to solve this LP is with generic solvers—an approach which is valid, but can be costly. One of our contributions is in identifying natural cases where min-budget contracts take on *simple* forms, which are easier to optimize, and have practical merit. In particular, we show that binary-action contracts have *all-or-nothing* structure ($t(j) \in \{0, B\}$), and plausible structural assumptions on the learning curve give rise to *threshold* contracts ($t(j) = B\mathbb{1}[j \geq j_0]$ for some $j_0$). Our theoretical results are summarized in Table 1, and detailed in the rest of the section.

## 3.3 All-or-nothing contracts: Binary action space and the statistical connection

We begin with a simple delegated learning setting in which the agent can choose one of two actions, $\mathcal{A} = \{n_1, n_2\}$, e.g., a 'small' vs. 'large' training set, and the principal seeks to incentivize training with more data.[4] This reduced case will be useful as a building block for the general case (which we return to in Sec. 3.4), and for making a precise connection between contract design and hypothesis tests.

**Simple min-budget contracts for binary action space.** Our first result shows that optimal binary-action contracts are all-or-nothing contracts whose budget is determined by the *total variation distance* between outcome distributions $f_2$ and $f_1$, namely $\|f_2 - f_1\|_{\text{TV}} = \frac{1}{2}\sum_{j=0}^m |f_{2,j} - f_{1,j}|$.

**Theorem 1** (Optimal binary-action contract). *In a binary-action contract setting with outcome distributions $f_1, f_2$ and costs $c_1, c_2$, the min-budget contract is an all-or-nothing contract, given by:*

$$t^*(j) = B\mathbb{1}[f_2(j) \geq f_1(j)] \quad \forall j \in \{0,\dots,m\}, \quad \text{where} \quad B = (c_2 - c_1)/\|f_2 - f_1\|_{\text{TV}}. \tag{5}$$

The proof (in Appendix B.4.2) is by LP duality. Intuitively, the optimal contract pays the agent for outcomes that are more likely to come from $f_2$ than from $f_1$. Moreover, it requires a higher budget the smaller the distance is between the two distributions $f_1, f_2$. At the extremes, if their distance is 1 (i.e. no overlap among their supports), the required budget for incentivizing $n_2$ is $c_2 - c_1$, whereas if their distance is 0 (i.e. $f_1 = f_2$) it becomes impossible to incentivize $n_2$.

---

[3]The MIN-BUDGET LP is closely related to the well-known MIN-PAY LP from non-budgeted contract design [e.g., 19], but with a different objective, and hence very different optimal solutions (see Appx. B.3).

[4]When $n_2$ is not the target or when it is not implementable, then the solution is immediate: always pay $c_1$.

**Formal connection to optimal hypothesis testing.** Theorem 1 also uncovers a direct correspondence between optimal contracts and optimal hypothesis tests. Intuitively, given the outcome distributions $\{f_1, f_2\}$, and in order to incentivize $n_2$, the principal wishes to pay the agent if the observed outcome $j \in \{0, \ldots, m\}$ is more likely to have originated from $f_2$. The principal can attempt to identify whether the outcome $j$ is drawn from distribution $f_1$ or $f_2$ through hypothesis testing, where a hypothesis test $\psi : \{0, \ldots, m\} \to \{0, 1\}$ maps a sample $j$ to either $f_2$ (indicated by 1) or to the null hypothesis $f_1$ (indicated by 0). For our purpose it is convenient to allow tests to be non-integral, in which case $\psi : \{0, \ldots, m\} \to [0, 1]$ maps $j$ to a *probability* with which it originates from $f_2$. The quality of a hypothesis test is measured by summing its type-1 and type-2 errors: $\sum_{j=0}^{m} f_{1,j} \psi_j + \sum_{j=0}^{m} f_{2,j}(1 - \psi_j)$. The test that minimizes this sum is known as the *most powerful* hypothesis test, and has been characterized by Neyman and Pearson [45, 4.3].

We now turn to formally establishing the connection. For a fixed $B$, observe that every contract with budget $B$ can be mapped to a hypothesis test via the bijection $\psi(j) = {}^{t(j)}\!/_B$. Then:

**Theorem 2** (Optimal contract vs. test). *Consider binary-action contract design with distributions $f_1, f_2$ and costs $c_1, c_2$. A contract $t$ with budget $B$ is optimal if and only if its corresponding hypothesis test $\psi = {}^t\!/_B$ is maximum power with type-1 and type-2 errors summing to $1 - \frac{c_2 - c_1}{B}$.*

The proof (Appendix B.4.2) is by a non-linear variable transformation to the MIN-BUDGET LP. Theorem 2 implies that the optimal contract for the binary-action case (Theorem 1) is equivalent to the well-known Neyman-Pearson lemma characterizing the most powerful hypothesis test:

**Lemma 1** (Neyman-Pearson [e.g., 45]). *Let $f_1, f_2$ be two discrete probability distributions. Then the most powerful hypothesis test for $f_1, f_2$ is the likelihood ratio test $\psi(j) = \mathbb{1}\left[f_2(j) \geq f_1(j)\right]$, which attains the optimal bound $1 - \|p - q\|_{\mathrm{TV}}$ on the sum of type-1 and type-2 errors.*

Theorem 2 establishes a new formal connection between the two domains of contract design and hypothesis testing. In the context of contract design, it provides a statistical interpretation: A min-budget contract can be interpreted as an optimal hypothesis test, and the ability to distinguish between the two hypotheses determines the required budget. In the converse direction, it enables a new proof for the Neyman-Pearson using the min-budget contract given by Theorem 1 (see Appendix B.4.2).

### 3.4 All-or-nothing contracts: Beyond binary action

For general action spaces, optimal contracts are not guaranteed to be all-or-nothing. In fact, we show that determining whether there exists an all-or-nothing contract that is optimal is NP-hard:

**Theorem 3** (Hardness). *Finding a min-budget all-or-nothing contract is NP-hard.*

The proof is by reduction from 3SAT, and appears in Appx. B.5.2. Nonetheless, there are special but important cases—notably the binary outcome case[5]— in which results from Sec. 3.3 hold, suggesting that ideas from Thm. 1 apply more broadly. The following algorithm makes use of these ideas, showing good empirical performance, and provable performance under an MLRP condition (Sec. 3.5).

**Single binding action algorithm.** Revisiting the closed-form all-or-nothing contract in Eq. (5), we observe that the result is based on the fact that binary action spaces have only one alternative action. Building upon this observation, we propose the *single binding action* (SBA) algorithm, which computes a solution to Eq. (4) in the general (many-actions) case: Given target action $n^* \in \mathcal{A}$, loop over all actions $n \neq n^*$, and apply the closed-form formula in Eq. (5) to obtain an (all-or-nothing) contract $t^*(n, n^*)$. If the agent's best response (Eq. (2)) satisfies $n(t^*(n, n^*)) = n^*$, return $t^*$. If the loop ends without returning a contract, return 'fail'. The following claim shows the algorithm is sound:

**Proposition 2** (Soundness of SBA). *When the single binding action algorithm terminates successfully, it returns an optimal contract which is an all-or-nothing contract.*

Proof in Appendix B.6. Prop. 2 ensures that if SBA succeeds, then the returned all-or-nothing contract $t^*$ is optimal. Moreover, failure does not preclude the existence of an optimal all-or-nothing contract. If SBA fails, we solve Eq. (4) with a generic LP solver. Empirically, SBA was successful in more than 85% of cases, and is $\sim\!10^3$ times faster than the LP solver (Appendix C.3). Thus, this optimistic 'try SBA first' approach typically succeeds, adds negligible overhead if not, and guarantees correctness.

---

[5]In this case there are *binary outcomes* ($m = 1$), such as when the agent's efforts can result in either success or failure [6, 28, 21]. For this case, we show that the optimal contract is also all-or-nothing (Appx. B.5.1).

## 3.5 Threshold contracts: MLRP assumption

In this section we provide sufficient conditions, in the form of natural structural properties of (stochastic) learning curves, that guarantee the optimality of even *simpler* contracts—namely *threshold contracts*—and the success of SBA. With inspiration from hypothesis testing [40] and contract theory [26, 19], it is natural to consider the *monotone likelihood ratio property* (MLRP) assumption:

**Definition 1** (MLRP [e.g., 26]). *A contract design setting satisfies MLRP if for every pair of actions $a, a'$ such that $c_a < c_{a'}$, the likelihood ratio $f_{a'}(j)/f_a(j)$ is monotonically increasing in $j$.*

In our context, MLRP states that the better the validation-set performance of a classifier, the more likely it was trained on more data. This holds in particular for monotone learning curves with binomial outcome distribution [19, B.1]. For a binary action space, MLRP ensures that $n_2$ is always implementable,[6] and that the optimal contract is a threshold contract: $t^*(j) = B\mathbb{1}[j \geq j_0]$. This is by Theorem 1, and by the fact that $\exists j_0$ such that $f_2(j)/f_1(j) \geq 1$ iff $j \geq j_0$ (see also Appendix B.7.1).[7] Interestingly, this is similar to the relation between the Neyman-Pearson lemma and the Karlin-Rubin theorem, which characterizes the most powerful hypothesis test under monotone likelihood ratio [40].

**MLRP for general action space.** MLRP does not guarantee threshold contracts in general: In Appendix B.7.3, we give a constructive counterexample satisfying MLRP, but for which the optimal contract is not threshold. However, refining MLRP to also capture 'diminishing returns' turns out to be sufficient for recovering guarantees generally. For target action $a_N$, denote the *survival probability* of an action $a_i$ by $s_i = \mathbb{P}_{j \sim f_i}[j \geq j^*]$, where $j^*$ is the minimal outcome at which action $a_N$ is more likely than $a_{N-1}$. These will serve as formal means for capturing the concavity of learning curves.

**Definition 2** (C-MLRP). *A contract design setting satisfies Concave-MLRP (C-MLRP) if it satisfies MLRP, and additionally the actions' survival probability is concave as a function of the actions' cost.*

Our final result shows that C-MLRP guarantees optimality of threshold contracts, and success of SBA:

**Theorem 4** (Sufficiency for threshold). *Consider a contract design setting with C-MLRP. Then the optimal contract is a threshold contract, and is recovered by the SBA algorithm.*

We prove this claim by showing that concavity implies that only one alternative action is binding in the linear program equivalent to Eq. (4), reducing the problem to the two-action case. By applying Theorem 1, we obtain optimality of threshold contracts in this case as well (proof in Appendix B.7.3). This also implies that SBC always terminates successfully on inputs that satisfy C-MLRP.

In practice, we believe that C-MLRP is a reasonable assumption for realistic learning curves: in Appendix B.8.1, we prove it is satisfied by a standard theoretical model of learning curves, and in Appendix C.3, we empirically demonstrate that it is (approximately) satisfied in settings where threshold contracts are optimal. Interestingly, in our empirical study, threshold contracts were often optimal even when C-MLRP did not hold—suggesting the condition is sufficient, but not necessary (see Sec. 4.1).

## 4 Experiments

We now turn to our empirical investigation of delegated learning under full and partial information. We base our experiments on the recently curated Learning Curves Database (LCDB) [43], which includes a large collection of stochastic learning curves for multiple classification datasets and methods. For each dataset and method, the database includes held-out accuracy measurements obtained for increasing sample sizes $n \in \{2^4, 2^{4.5}, \ldots, 2^{15}\}$, with multiple repetitions per $n$; these provide us with stochastic learning curves. Here we focus primarily on the popular MNIST dataset [39] as our case study, and on MLP and GBDT as representative classifiers, but we refer the reader to Appendix C for further experiments on additional datasets and methods. Code is available at: https://github.com/edensaig/delegated-classification.

### 4.1 Full information

We begin with the full information setting to explore in a clean environment how different parameters of the learning setting and environment affect predictive performance and economic outcomes.

---

[6]As a corollary of a similar result for min-pay contracts [19, Lemma 7].

[7]This also holds for binary outcomes in an arbitrary action space, see Appendix B.7.2.

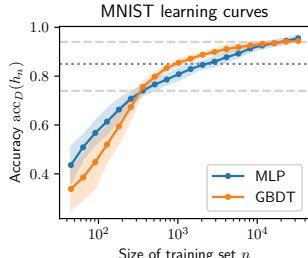 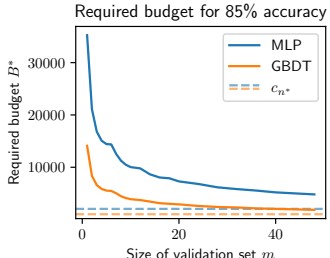 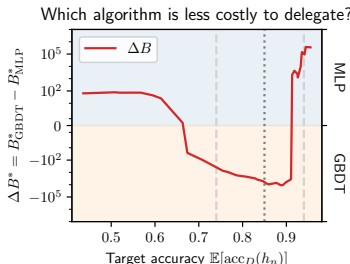

Figure 3: Delegating with full information. **(Left)** Typical learning curves for two learning algorithms on MNIST. **(Center)** Required budget for target accuracy of $0.85$ per validation set size $m$. **(Right)** Different cost regimes, indicating per accuracy region which of the two methods is cheaper to delegate.

**Validation set size.** Fig. 3 (left) presents typical stochastic learning curves for two learning algorithms: Multi-Layered Perceptron (MLP) and Gradient-Boosted Decision Trees (GBDT). We take an arbitrary accuracy point on the curve at $\mathrm{acc}(n) = 0.85$ (dotted line) to examine the effects of validation set size $m$ on min-budget contracts. Notice that MLP requires larger $n$ to obtain $0.85$; Fig. 3 (center) shows how this translates to a larger required budget $B^*$, which holds for all $m$. As $m$ increases, required budgets and the difference between them both decrease. Larger validation sets are therefore useful for reducing required budget. Nonetheless, even for reasonable $m$, obtained budgets still remain higher than their theoretical lower bounds (target action costs $c_{n^*}$).

**Budget regimes.** Fig. 3 (left) also indicates two points in which the learning curves cross (dashed lines), at $\sim 0.74$ and $\sim 0.94$ accuracy. These correspond to sample sizes $n$ for which both methods obtain matching accuracies (in expectation). For a self-sufficient learner, the implication is that at each of these points, both methods are equally costly, i.e., both cost $c_n$. Interestingly, and in contrast, delegation can entail different required budgets *despite* equal accuracies. Fig. 3 (right) shows for each target accuracy the gap in required budgets between both methods, $\Delta B^* = B^*_{\mathrm{GBDT}} - B^*_{\mathrm{MLP}}$. As can be seen, each method is comparatively more (or less) costly in different accuracy regimes (up to 0.6; between 0.6 and 0.92; and above 0.92). Crucially, the budget gap can be large even when accuracies match (dashed lines). For example, even though both MLP and GBDT require $n=362 \approx 2^{17/2}$ samples to obtain $\sim 0.74$ accuracy, GBDT is cheaper ($\Delta B^* = -10^2$); for $\sim 0.94$ which requires $n=23170 \approx 2^{29/2}$ from both, GBDT is significantly more expensive ($\Delta B^* = 10^5$). The reason for this is that optimal budgets are determined by the ability to distinguish between distributions (Sec. 3.3).

**Prevalence of simple contracts.** To understand the applicability of our theoretical findings, in Appendix C.3 we conduct an empirical prevalence evaluation on additional learning algorithms, and across target actions. We observed that min-budget contracts assume a threshold form and the SBC algorithm returns correct results in more than 85% of cases overall. Restricting optimization to simple contracts, budget requirements were generally less than 1% higher than that of a min-budget contract, suggesting that simple contracts may provide a good approximation even when min-budget contracts do not assume a simple form.

## 4.2 Partial information

We now turn to consider delegation under partial information, in which the principal must rely on an estimated learning curve. We instantiate this idea by assuming that the principal has access to a small 'pilot' dataset of size $k$, where $k$ is considered small. Using this set, the principal creates an estimated learning curve $\hat{F}$ by fitting a curve to accuracies obtained for up to some $n_0 \le k$, and extrapolating to larger $n > n_0$. In particular, we experiment with fitting parametric power-law curves of the form $\mathbb{E}[\alpha_n] = a - bn^{-c}$, which have been shown to provide good fit in various scenarios both empirically and theoretically [49, 34, 11]. Since power-law curves are monotone, composition with binomial distributions increasing in $p$ provably results in MLRP stochastic curves [19, B.1].

**Bias-variance tradeoff.** Given $k$ pilot examples, there are different ways in which the principal can use them to construct an estimated curve. Here we consider a simple tradeoff: setting $n_0$ to be small but with more samples per $n < n_0$ (low variance), or setting $n_0$ to be large but with few samples per $n < n_0$ (low bias). We define $r$ as the number of samples per $n$ (so low $r$ means larger $n_0$). Then, for a

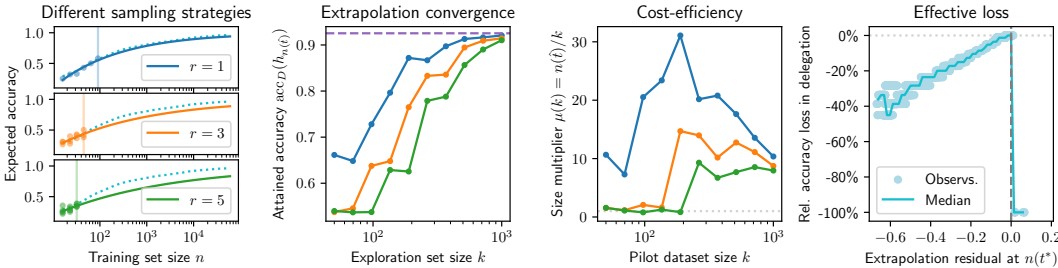

Figure 4: Delegating with partial information. **(Left)** Extrapolated learning curves for different $r$. **(Center-left)** Accuracy obtained via delegation per pilot set size $k$. **(Center-right)** Multiplicative gain in effective number of samples due to delegation. **(Right)** Implications of over vs. under-estimation.

given $r$, we set $n_0$ such that $\sum_{n \leq n_0} r \cdot n \leq k$ (i.e., such that the total number of used samples does not exceed $k$). Fig. 4 (left) shows different curve fits for $r \in \{1, 3, 5\}$, and corresponding $n_0$. Then, Fig. 4 (center-left) shows for a certain fixed budget the accuracy level that can be attained for increasing $k$, and as a function of $r$. As can be seen, having sufficient points $k$ for constructing $\hat{F}$ is important, but performance grows quickly with $k$ (note log-scale x-axis). It is also apparent in our example that low bias (via larger $n_0$) is much more important than low variance for constructing useful $\hat{F}$.

**Cost-efficiency tradeoff.** Because the pilot set provides the principal a basic means for obtaining minimal accuracy, we can ask: given $k$ examples, and for a fixed budget $B$, what is the added benefit of delegating learning? For this, we define $\mu(k) = n(\hat{t})/k$ to be the *sample-size multiplier*, i.e., the multiplicative gain in the effective number of samples due to delegation. Fig. 4 (center-right) shows $\mu(k)$ for increasing $k$ and across $r$. For $r = 1$ (which is superior in terms of performance and outcomes), $\mu$ begins at $\sim 10$, increases to $\sim 30$ at around $k = 190$, and slowly decreases back to $\sim 10$ towards $k = 1,000$. For $r > 1$, we observe that $\mu \approx 1$, i.e., there is effectively no gain from delegation, until around $k = 100$, only after which some gain is restored. This highlights the importance of obtaining an accurate estimate $\hat{F}$ in terms of the economic consequences of delegation.

**Over vs. under-estimation.** Typically in curve-fitting, over and under-estimation are treated equally, since both types of error can negatively affect goodness of fit and extrapolation quality. However, for delegation, the implications of over vs. under-estimation on contract outcomes are highly asymmetric. Fig. 4 (right) shows for a target incentivized number of samples $n(t^*)$ the relation between the (theoretical) *signed* extrapolation error $n(t^*)$ (i.e., over- or under-estimate, measured in accuracy points) and the eventual loss in accuracy obtained through delegation, relative to perfect estimation. Each point in the plot corresponds to one curve-fitting instance, with points shown for varying $k$, $n_0$, and $r$, and with multiple independent repetitions. Results show that in the under-estimation regime (i.e., *negative* extrapolation error), loss in accuracy degrades gracefully with the estimation error. In stark contrast, even minimal over-estimation (*positive* extrapolation error) causes accuracy to plummet dramatically, as the agent's rational response in those cases was to use the smallest dataset possible. We interpret this as a consequence of 'setting the bar too high'—a rational decision to minimize effort in response to unrealistic expectations. This has important implications for the choice of how to fit and extrapolate learning curves, suggesting that contracts can be tolerant to under-estimation, while over-estimation should be avoided at all costs.

## 5 Discussion

Motivated by the increasingly-common practice of outsourcing learning tasks, this paper sets out to introduce and study the novel problem of delegated classification. Our findings suggest that conflict of interests should not be overlooked, and that contracts hold potential as a means for aligning them. Our analysis relies on a set of assumptions, which should be carefully considered by practitioners and empiricists alike; we also believe that there are likely further fruitful connections to explore between contracts and statistical hypothesis testing. As a problem of contract design, and when the learning task is reasonably well-behaved, delegated learning manifests in the form simple threshold contracts. A natural question for future work is whether simplicity also implies *robustness* to partial knowledge—as is often the case [19].

**Acknowledgements.** The authors would like to thank Ruth Heller, Shafi Goldwasser, Jonathan Shafer, Ohad Einav, and anonymous reviewers for their insightful remarks and valuable suggestions. Nir Rosenfeld is supported by the Israel Science Foundation grant no. 278/22. Eden Saig is supported by the Israel Council for Higher Education PBC scholarship for Ph.D. students in data science. Funded by the European Union (ERC, ALGOCONTRACT, 101077862, PI: Inbal Talgam-Cohen).

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

## A   Broader implications

In this paper, we set out to formalize and study the task of delegating classification through the lens of contract design. Given that our work is largely motivated by the increasingly common practice of outsourcing learning tasks to specialized service providers, we believe our algorithm, analysis, and empirical observations carry meaningful implications for practitioners and decision-makers alike. At the same time, it is important to remember that our work—as others considering economic aspects of learning—studies delegation in a simplified setting and under certain assumptions. As such, and since devising and agreeing to legally-binding contracts can have concrete implications on real-world outcomes, care should be taken when applying ideas or conclusions that derive from our work in practice.

For example, consider that our formalism relies on the assumption that the learning agent is all-knowing and rational. Yet in reality, agents must act under partial information, face irreducible (and often unquantifiable) uncertainty, and—being human—are subject to common behavioral biases. It is unclear a-priori if and how our statements and conclusions carry over to this setting. As another example, notice that our formalism considers a one-shot setting where a single contract between a single principal-agent pair is instantiated once. But in reality, competition and long-term reputation may play a significant role in determining the agent's incentive structure, and consequently, her behavior. In such cases, naïvely applying our framework without careful inspection of the appropriate incentives on both ends can result in sub-optimal contracts, possibly to the detriment of all involved parties. Nonetheless, given that our work aims to take an initial step towards establishing delegated learning, we view its extension to non-rational agents and temporal and competitive settings as intriguing future work.

One message that our work conveys is that in delegation, simplicity, in the form of threshold contracts, has merit. This draws connections to other works that similarly argue for simplicity as an important and useful property of effective delegation mechanisms. Our work shows that simple contracts are, under reasonable conditions, computationally feasible and theoretically optimal. Economically, threshold contracts are practically appealing since they are easy to understand, communicate, and regulate. Given that contracts are in essence social constructs, we believe these properties are key for establishing threshold contracts as effective building blocks for machine learning markets.

## B   Min-budget contract design – deferred proofs

**Notation.**   To align with traditional contract design notation, in this section, we denote the action space by $\mathcal{A} = [n] = \{1, \ldots, n\}$, and the outcome space by $\Omega = \{0, \ldots, m\}$. We denote by $\Delta(\mathcal{X})$ the set of distributions over a given set $\mathcal{X}$. For contract design problems, we denote the outcome probabilities by $F_{i,j}$, such that $F_i \in \Delta(\Omega)$ is the outcome distribution associated with action $i$, and also denote the cost of each action by $c_i$. Given $x \in \mathbb{R}$, we denote $x^+ = \mathrm{ReLU}(x) = \max\{0, x\}$. We denote the indicator function by $\mathbb{1}[\cdot]$, the total variation distance between distributions $P, Q$ by $\|P - Q\|_{\mathrm{TV}}$ (see Definition 4), and the survival function of $P \in \Delta(\Omega)$ by $\mathbb{S}_P(\cdot)$ (see Definition 14).

**A note on individual rationality.**   In the contract design literature, a contract is said to be *incentive compatible* (IC) with respect to some action $a^*$ if it satisfies $u_{a^*}(t) \geq u_a(t)$ for all actions $a \in \mathcal{A}$. In the delegated classification setting, this corresponds to the constraint $n(t) = n^*$ in Eq. (4). As an additional constraint, contracts in which the agent's expected utility is always non-negative ($u_{n^*}(t) \geq 0$) are said to be *individually rational* (IR). In the case of delegated classification, it is natural to assume that there always exists a valid action that the agent can take at zero cost — for example, returning a dummy classifier that always abstains from prediction (thus having zero accuracy), or a classifier which makes a prediction at random. As such, contracts in the delegated classification setting can be assumed to be individually rational without loss of generality, as any action $n$ that is chosen by the agent has utility which is weakly larger than the utility of the zero-cost action, which is always non-negative. Moreover, even in cases where a zero-loss action does not exist, we show that individual rationality (IR) can be attained in a straightforward manner, by adding the minimal cost $c_1$ to each entry of an incentive compatible (IC) contract ($t_j \mapsto t_j + c_1$):

**Claim 1.** *Given an IC contract $t$ with $c_1 > 0$, the contract $t + c_1$ (add $c_1$ to every coordinate of the contract) is both IC and IR.*

*Proof.* An IC contract implementing action $a^*$ satisfies $u_{a^*}(t) \geq u_a(t)$, for all actions $a \in \mathcal{A}$. In particular, this inequality holds in relation to the least-costly action, denoted by $1 \in \mathcal{A}$. Plugging the definition of the agent's expected utility (Eq. (2)), we obtain $u_{a^*}(t) \geq \mathbb{E}_{j \sim f_1}[t] - c_1$. Thus, under

the mapping $t'_j = t_j + c_1$, it holds that:

$$u_{a^*}(t') = u_{a^*}(t + c_1) \geq \mathbb{E}_{j \sim f_1}[t + c_1] - c_1 = \mathbb{E}_{j \sim f_1}[t] \geq 0$$

and therefore the contract $t' = t + c_1$ is individually rational and still implements action $a^*$. $\square$

### B.1 Relation between budget-optimal and min-budget contracts

*Proof of Proposition 1.* Given budget $B > 0$, denote by $t_{\text{BO}}$ the budget-optimal contract, and denote the action it implements by $n^*$. By definition, $t_{\text{BO}}$ is a feasible solution to the min-budget contract design problem implementing action $n^*$. Denote by $t_{\text{MB}}$ the corresponding optimal solution to the min-budget problem implementing action $n^*$ (Eq. (4)). $t_{\text{MB}}$ implements the same action $n^*$ by definition, and satisfies $\|t_{\text{MB}}\|_\infty \leq \|t_{\text{BO}}\|_\infty \leq B$ due to the optimization objective. Hence, $t_{\text{MB}}$ is a budget-optimal contract for the given budget $B$, which is also min-budget. $\square$

**Iterative min-budget.** To find the budget-optimal contract using iterative applications of Eq. (4), we observe that any budget-limited contract $t : \Omega \to [0, B]$ has bounded expected pay $\mathbb{E}_{f_n}[t] \leq B$ for any distribution $f_n$. Hence, actions $n'$ with cost $c_{n'} > B$ cannot be implemented using budget $B$, as the agent's utility $u_{n'} = \mathbb{E}_{f_{n'}}[t] - c_{n'} < 0$ will be smaller than the utility of the zero-cost action $u_0 = \mathbb{E}_{f_n}[t] - 0 \geq 0$. Define the reduced action set:

$$\mathcal{A}' = \{n \in \mathcal{A} \mid c_n \leq B \wedge n \text{ is implementable}\}$$

$\mathcal{A}'$ is finite when $\mathcal{A}$ is finite, or when the data cost is unbounded and the learning curve is monotone. To find $n^*$ within this space, go over all $n \in \mathcal{A}'$, calculate the minimal budget $B_n$ required for implementation, and take $\operatorname{argmin}_{n \in \mathcal{A}'} B_n$. This is possible within a finite number of steps.

### B.2 The min-budget linear program and equivalent forms

The min-budget contract (Eq. (4)) implementing action $i \in \mathcal{A}$ is given by the MIN-BUDGET linear program:

$$
\begin{aligned}
\min_{t \in \mathbb{R}_{\geq 0}^{|\Omega|}, B \in \mathbb{R}_{\geq 0}} \quad & B \\
\text{s.t.} \quad & \\
\forall j \in \Omega : \; & t_j \leq B \qquad\qquad\qquad\quad \text{(BUDGET)} \\
\forall i' \neq i : \; & \sum_{j \in \Omega} F_{i',j} t_j - c_{i'} \leq \sum_{j \in \Omega} F_{i,j} t_j - c_i \quad \text{(IC)}
\end{aligned}
\tag{6}
$$

The dual of the min-budget LP is given by:

**Claim 2.** *The dual linear program of Eq. (6) is given by:*

$$
\begin{aligned}
\max_{\lambda \in \mathbb{R}_{\geq 0}^{n-1}, \mu \in \mathbb{R}_{\geq 0}^{|\Omega|}} \quad & \sum_{i' \neq i} (c_i - c_{i'}) \lambda_{i'} \\
\text{s.t.} \quad & \\
\forall j \in \Omega : \; & \sum_{i' \neq i} (F_{i,j} - F_{i',j}) \lambda_{i'} \leq \mu_j \\
& \sum_{j \in \Omega} \mu_j \leq 1
\end{aligned}
\tag{7}
$$

*Proof.* We take the dual by translating the optimization problem into canonical form. The canonical form we target:

$$\min_{x \geq 0} c^T x \quad \text{s.t.} \quad Cx \leq d$$

and its dual, as given by Lahaie [38], is:

$$\max_{y \geq 0} -d^T y \quad \text{s.t.} \quad C^T y \geq -c$$

To translate Eq. (6), we note that in our case the components $c, C, d$ are given by:

$$c^T = \begin{pmatrix} 0 & \cdots & 0 & 1 \end{pmatrix} \in \mathbb{R}^{|\Omega|+1}$$

$$C = \left( \begin{array}{ccccc|c} F_{1,0} - F_{i,0} & \cdots & & F_{1,|\Omega|} - F_{i,|\Omega|} & & 0 \\ \vdots & \ddots & & \vdots & & \vdots \\ & & F_{i',j} - F_{i,j} & & & \\ & & & \ddots & & \\ F_{n,0} - F_{i,0} & & & F_{n,m} - F_{i,m} & & 0 \\ \hline & & & & & -1 \\ & & I_{m+1} & & & \vdots \\ & & & & & -1 \end{array} \right) \in \mathbb{R}^{(n-1+|\Omega|) \times (|\Omega|+1)}$$

$$d^T = \begin{pmatrix} c_1 - c_i & \cdots & c_n - c_i & | & 0 & \cdots & 0 \end{pmatrix} \in \mathbb{R}^{n-1+|\Omega|}$$

To simplify formulation, we denote the dual optimization variable $y$ as follows:

$$y^T = \begin{pmatrix} \lambda_1 & \cdots & \lambda_{i-1} & \lambda_{i+1} & \ldots \lambda_n & | & \mu_0 & \cdots & \mu_m \end{pmatrix} \in \mathbb{R}^{n-1+|\Omega|}$$

Under this formulation, the dual's objective is:

$$-d^T y = \sum_{i \neq i'} (c_i - c_{i'}) \lambda_{i'} \tag{8}$$

The constraints given by the first $m$ rows of $C^T$ are:

$$\forall j \in \Omega : \sum_{i' \neq i} (F_{i',j} - F_{i,j}) + \mu_j \geq 0$$

and equivalently:

$$\forall j \in \Omega : \sum_{i' \neq i} (F_{i,j} - F_{i',j}) \leq \mu_j \tag{9}$$

The constraint corresponding to the last row of $C^T$ is given by:

$$\sum_{j \in \Omega} -\mu_m \geq -1$$

and equivalently:

$$\sum_{j \in \Omega} \mu_m \leq 1 \tag{10}$$

Combining equations (8, 9, 10) yields the linear program given by Eq. (7). $\qquad\square$

To map between contracts and hypothesis tests, we introduce the following variable transformation:

**Definition 3** (Statistical representation of contracts). *For a given contract* $t : \Omega \to \mathbb{R}_{\geq 0}$, *denote* $B = \max_j t_j$. *The statistical representation of* $t$ *is given by:*

$$t_j = \phi_j \beta^{-1}$$

*where* $\phi_j \in [0, 1]$ *and* $\beta = B^{-1}$.

Note that the transformation $(t, B) \mapsto (\phi, \beta)$ is non-linear, and well-defined for all $B > 0$. Under this variable transformation, the MIN-BUDGET transforms to an equivalent linear program:

**Lemma 2.** *For a feasible design problem, the min-budget linear program (Eq. (6)) is equivalent to:*

$$\max_{\phi \in [0,1]^{|\Omega|}, \beta \in \mathbb{R}_{\geq 0}} \beta$$

$$\text{s.t.} \tag{11}$$

$$\forall i' \neq i : \sum_{j \in \Omega} F_{i,j}(1 - \phi_j) + \sum_{j \in \Omega} F_{i',j}\phi_j \leq 1 - (c_i - c_{i'})\beta$$

*Proof.* Given Eq. (6), define $\phi \in [0,1]^{|\Omega|}$, $\beta \geq 0$ according to Definition 3:

$$B = \beta^{-1}$$
$$t_j = \phi_j B = \phi_j \beta^{-1} \tag{12}$$

Under the transformation defined by Eq. (12), the (BUDGET) constraint in Eq. (6) transforms as follows:

$$t_j \leq B \Leftrightarrow \phi_j \beta^{-1} \leq \beta^{-1}$$
$$\Leftrightarrow \phi_j \leq 1 \tag{13}$$

and the (IC) constraint in Eq. (6) transforms as:

$$\sum_j F_{i,j}t_j - c_i \geq \sum_j F_{i',j}t_j - c_{i'} \Leftrightarrow \sum_j (F_{i,j} - F_{i',j})\phi_j \beta^{-1} \geq c_i - c_{i'}$$

$$\Leftrightarrow \sum_j F_{i,j}\phi_j - \sum_j F_{i',j}\phi_j \geq (c_i - c_{i'})\beta$$

$$\Leftrightarrow 1 - \sum_j F_{i,j}(1 - \phi_j) - \sum_j F_{i',j}\phi_j \geq (c_i - c_{i'})\beta \tag{14}$$

$$\Leftrightarrow \sum_j F_{i,j}(1 - \phi_j) + \sum_j F_{i',j}\phi_j \leq 1 - (c_i - c_{i'})\beta$$

where the first equivalence is by Eq. (12), and the third equivalence is valid as $\sum_j F_{i',j} = 1$ for all $i'$. Finally, the objective of Eq. (6) transforms as:

$$\min B \Leftrightarrow \max \beta \tag{15}$$

Combining equations (13, 14, 15) yields the linear program in Eq. (11). □

### B.3 Relation to min-pay contract design

Min-pay contract design aims to design a contract which minimizes the expected pay under the implemented action $n^*$:

$$t^* = \text{argmin}_t \mathbb{E}_{j \sim f_{n^*}}[t_j] \quad \text{s.t.} \quad n(t) = n^* \tag{16}$$

In contrast to the min-budget contract (Eq. (4)), the $\|t\|_\infty$ objective measuring maximal pay is replaced with the $\mathbb{E}_{j \sim f_{n^*}}[t_j]$ objective measuring expected pay. Eq. (16) is equivalent to the MIN-PAY linear program:

$$\min_{t \in \mathbb{R}_{\geq 0}^{|\Omega|}} \sum_{j \in \Omega} F_{i,j}t_j$$

$$\text{s.t.} \tag{17}$$

$$\forall i' \neq i : \sum_{j \in \Omega} F_{i',j}t_j - c_{i'} \leq \sum_{j \in \Omega} F_{i,j}t_j - c_i \quad \text{(IC)}$$

#### B.3.1 Equivalence of implementability

The implementatbility of min-pay contracts is characterized in Dütting et al. [19]. We cite the main result for completeness:

**Proposition 3** (Min-pay implementability; [19], A.2)**.** *An action $i \in \mathcal{A}$ is implementable (up to tie-breaking) if and only if there is no convex combination of the other actions that results in the same distribution $f_i = \sum_{i' \neq i} \alpha_{i'} f_{i'}$, but lower cost $c_i > \sum_{i' \neq i} \alpha_{i'} c_{i'}$.*

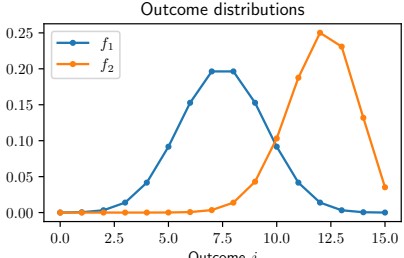
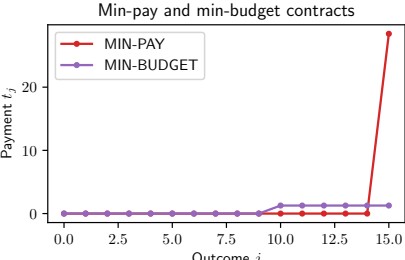

Figure 5: Qualitative comparison of min-pay and min-budget contracts under MLRP. **(Left)** The contract design setting, representing two possible actions with binomial outcome distributions ($p_1 = 0.5$, $p_2 = 0.8$, $m = 15$). Action 1 has zero cost $c_1 = 0$, and action 2 has unit cost $c_2 = 1$. **(Right)** The resulting min-pay (red) and min-budget (purple) contracts, obtained by solving Eq. (17) and Eq. (6), respectively. The min-pay awards payment only for the highest outcome, in alignment with Proposition 4; the min-budget is a threshold contract, in alignment with claim 12.

We leverage to Proposition 3 to characterize the implementability of min-budget contracts. This is possible due to the following connection:

**Claim 3** (Implementability equivalence). *A contract $t$ is feasible solution of MIN-BUDGET if and only if it is a feasible solution of MIN-PAY.*

*Proof.* A contract $t$ which satisfies MIN-BUDGET (Eq. (6)) satisfies the (IC) constraint, and thus also satisfies MIN-PAY (Eq. (17)). Conversely, a contract $t$ which satisfies MIN-PAY satisfies the (IC) constraint. Set $B = \max_j t_j$ and obtain a feasible solution to MIN-BUDGET. □

### B.3.2 Functional form of min-pay contracts under MLRP

The optimal min-pay contracts under the MLRP assumption (Definition 1) are characterized in [19]:

**Proposition 4** (Optimal min-pay contract under MLRP; [19], Lemma 7). *Consider a contract design setting for which MLRP holds. If the highest-cost action $n$ is implementable, then there is a min-pay contract has a single nonzero-payment, which is rewarded for the highest outcome $m$.*

See Fig. 5 for a qualitative comparison of min-pay and min-budget contracts.

**Min-pay contracts in delegated classification settings with MLRP.** While simple, the min-pay contract given by Proposition 4 would be impractical in many realistic scenarios of delegated classification. In a delegated classification setting, the highest outcome corresponds to 100% validation set accuracy (i.e all validation set samples classified correctly). As $m$ grows, the highest outcome becomes exponentially less likely, and the min-pay contract awards increasingly high payment with increasingly low probability (See Fig. 6). In such settings, even a slight degree of risk-aversion is likely to affect the agent's decisions: From the agent's perspective, the probability of receiving any payment from a min-pay contract may become small to a degree where even a slight degree of agent risk-aversion will manifest itself in the decision-making process. From the principal's perspective, even though min-pay contracts guarantee small payment in expectation, the amount payment in the case of a rare event would quickly become unfeasible.

### B.4 Min-budget contracts with two actions

In this section, we explore min-budget settings with two actions, and prove Theorem 2. When $n = 2$, assume without loss of generality that the contract implements action $i = 2$, and denote $c = c_2 - c_1 > 0$.

We recall the definition of total variation distance:

**Definition 4** (Total variation distance). *Given two distributions $P, Q \in \Delta(\Omega)$, the total variation distance between $P$ and $Q$ is:*

$$\|P - Q\|_{\mathrm{TV}} = \frac{1}{2} \|P - Q\|_1$$

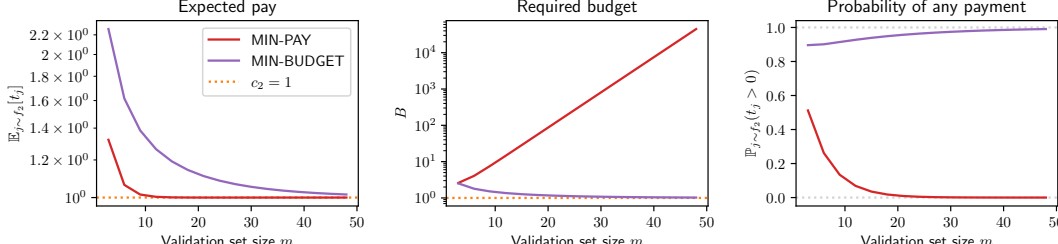

Figure 6: Comparison of min-pay and min-budget contracts under MLRP for varying validation set size $m$. The delegation setting is similar to the one depicted in Fig. 5: two binomial-outcome actions ($p_1 = 0.5$, $p_2 = 0.8$) and varying $m$. Action 1 has zero cost $c_1 = 0$, and action 2 has unit cost $c_2 = 1$. In the left and center plots, the cost of action 2 is represented by an orange line ($c_2 = 1$) representing the lower bound as in Fig. 3 (Center). **(Left)** Expected pay $\mathbb{E}_{j \sim f_2}[t_j]$. **(Center)** Required budget $\|t\|_\infty$. **(Right)** Probability of getting any payment $\mathbb{P}_{j \sim f_2}[t_j > 0]$.

The following equivalent definition is useful:

**Claim 4.** *Let $P, Q \in \Delta(\Omega)$. It holds:*

$$\|P - Q\|_{\text{TV}} = \sum_{j \in \Omega} (Q_j - P_j)^+$$

*where $x^+ = \max\{x, 0\}$.*

*Proof.* By definition:

$$\|P - Q\|_{\text{TV}} = \frac{1}{2} \|P - Q\|_1$$

Decompose the $L_1$ norm:

$$\frac{1}{2} \|P - Q\|_1 = \frac{1}{2} \sum_{j \in \Omega} \left( (Q_j - P_j)^+ + (P_j - Q_j)^+ \right)$$

As $\sum_{j \in \Omega} P_j = \sum_{j \in \Omega} Q_j = 1$, it holds that:

$$\sum_{j \in \Omega} (Q_j - P_j)^+ = \sum_{j \in \Omega} (P_j - Q_j)^+$$

and therefore $\|P - Q\|_{\text{TV}} = \sum_{j \in \Omega} (Q_j - P_j)^+$ as required. □

### B.4.1  Optimal two-action contract

**Theorem 5** (Two-action min-budget contract; formal statement of Theorem 1). *When $n = 2$, the optimal min-budget contract $t^*$ is given by:*

$$t_j^* = \frac{c}{\|F_2 - F_1\|_{\text{TV}}} \mathbb{1}\left[F_{2,j} \geq F_{1,j}\right] \tag{18}$$

*Proof.* We prove this claim using LP duality, by showing that the optimal primal objective corresponding to $t^*$ is identical to a feasible solution of the dual LP.

The primal LP (Eq. (6)) for two actions is given by:

$$\min_{t \in \mathbb{R}_{\geq 0}^{|\Omega|}, B \in \mathbb{R}_{\geq 0}} B \tag{19}$$

s.t.

$$\forall j \in \Omega : t_j \leq B \qquad \text{(BUDGET)}$$

$$\sum_{j \in \Omega} (F_{2,j} - F_{1,j}) t_j \geq c \qquad \text{(IC)}$$

As $t^*$ is bounded, the optimal objective $B^*$ corresponds to the maximal possible payout of $t^*$:

$$B^* = \max_{j \in \Omega} t_j^* = \frac{c}{\|F_2 - F_1\|_{\mathrm{TV}}} \tag{20}$$

and therefore the (BUDGET) constraint is satisfied. For the (IC) constraint, denote by $\Omega_\geq$ the following set:

$$\Omega_\geq = \{j \in \Omega \mid F_{2,j} \geq F_{1,2}\}$$

using this notation, we note that $t_j^* > 0$ if and only if $j \in \Omega_\geq$. Plugging into the constraint, we obtain:

$$\sum_{j \in \Omega} (F_{2,j} - F_{1,j})\, t_j^* = \sum_{j \in \Omega_\geq} (F_{2,j} - F_{1,j})\, B^*$$
$$= c\frac{\sum_{j \in \Omega_\geq} (F_{2,j} - F_{1,j})}{\|F_2 - F_1\|_{\mathrm{TV}}}$$
$$= c$$

Where $\|F_2 - F_1\|_{\mathrm{TV}} = \sum_{j \in \Omega_\geq} (F_{2,j} - F_{1,j})$ by definition. This shows that $t^*$ is a feasible solution for the primal LP.

To prove optimality, we show that the the dual linear program attains an identical objective. By Claim 2, the dual LP for two actions is given by:

$$\max_{\lambda \in \mathbb{R}_{\geq 0}, \mu \in \mathbb{R}_{\geq 0}^{|\Omega|}} c\lambda$$
$$\text{s.t.}$$
$$\forall j \in \Omega : (F_{2,j} - F_{1,j})\, \lambda \leq \mu_j \tag{21}$$
$$\sum_{j \in \Omega} \mu_j \leq 1$$

Denote the vector $\vec{v}_j(\lambda) \in \mathbb{R}_{\geq 0}^{|\Omega|}$:

$$\vec{v}_j(\lambda) = \lambda\, (F_{2,j} - F_{1,j})^+$$

We note that outcomes $j'$ for which $F_{2,j'} \leq F_{1,j'}$ (formally $j' \in \Omega \setminus \Omega_\geq$) correspond to constraints which are satisfied for any $\lambda \geq 0$. Otherwise $j \in \Omega_\geq$, and in these cases $\lambda$ can be increased until $\vec{v}(\lambda)$ saturates the simplex constraint $\sum_{j \in \Omega} \mu_j = 1$. The simplex constraint is binding for a value $\lambda^* > 0$ satisfying:

$$\sum_{j \in \Omega} \lambda^* (F_{2,j} - F_{1,j})^+ = 1$$

and therefore the optimal value $\lambda^*$ of the dual LP (Eq. (21)) is:

$$c\lambda^* = \frac{c}{\sum_{j \in \Omega} (F_{2,j} - F_{1,j})^+} = \frac{c}{\|F_2 - F_1\|_{\mathrm{TV}}} \tag{22}$$

Where the second equality is given by claim 4. The dual objective in Eq. (22) is identical to the primal objective attained by the contract $t^*$ in Eq. (20), and therefore $t^*$ is an optimal contract by strong LP duality. $\square$

### B.4.2 Contracts and hypothesis tests

We recall the formal definition of the Neyman-Pearson lemma:

**Lemma 3** (Neyman-Pearson, [e.g., 45, 4.3]). *Let $P, Q \in \Delta(\mathcal{X})$ be two probability measures over an arbitrary set $\mathcal{X}$. Then for any hypothesis test $\psi : \mathcal{X} \to \{0, 1\}$, it holds:*

$$P(\psi(x) = 1) + Q(\psi(x) = 0) \geq 1 - \|P - Q\|_{\mathrm{TV}}. \tag{23}$$

*Moreover, equality holds for the Likelihood Ratio test $\psi^*(x) = \mathbb{1}\,[q(x) \geq p(x)]$.*

In our analysis, we will assume that the space $\mathcal{X}$ is finite. We also note that Lemma 3 is stated for decision rules with binary output, but its domain can be extended to fractional decision functions $\psi : \mathcal{X} \to [0,1]$ without loss of generality: As the sum of errors is linear in $\psi$, the optimal fractional decision rule is a solution to the linear program $\min_{\psi \in [0,1]^{\mathcal{X}}} (P(\psi = 1) + Q(\psi = 0))$ where $P(\psi = 1) = \sum_{x \in \mathcal{X}} p_x \psi(x)$ and $Q(\psi = 0) = \sum_{x \in \mathcal{X}} q_x (1 - \psi(x))$. The feasible region of this linear program is the hypercube $[0,1]^{\mathcal{X}}$ and its vertices are the set of binary decision rules $\{0,1\}^{\mathcal{X}}$, which also includes the optimal binary rule $\psi^*$ given by Lemma 3. As every feasible linear program attains its optimum on a vertex, the binary $\psi^*$ is also the optimal among fractional rules.

*Proof of Theorem 2.* For a given two-action contract, denote $P_j = F_{1,j}$, $Q_j = F_{2,j}$, and $c = c_2 - c_1$. We recall that the statistical min-budget LP for two actions is given by Lemma 2:

$$\max_{\phi \in [0,1]^{|\Omega|}, \beta \in \mathbb{R}_{\geq 0}} \beta$$

$$\text{s.t.}$$

$$\sum_{j \in \Omega} P_j \phi_j + \sum_{j \in \Omega} Q_j (1 - \phi_j) \leq 1 - c\beta \tag{24}$$

and the Neyman-Pearson lemma is given by Eq. (23).

Given a min-budget contract design problem, apply Theorem 1 to obtain its optimal solution, as given by Eq. (18):

$$B^* = \frac{c}{\|P - Q\|_{\mathrm{TV}}}$$

$$t_j^* = \frac{c}{\|P - Q\|_{\mathrm{TV}}} \mathbb{1}\left[Q_j \geq P_j\right]$$

Applying the transformation from Definition 3 on the optimal contract, we obtain equivalently:

$$\beta^* = (B^*)^{-1} = \frac{\|P - Q\|_{\mathrm{TV}}}{c}$$

$$\phi_j^* = t_j^*/B^* = \mathbb{1}\left[Q_j \geq P_j\right]$$

Note that $\phi_j^*$ is a maximum-likelihood decision rule, similar to the optimal critical function in the Neyman-Pearson lemma. Additionally, by optimality of the contract-design optimization variable $\beta^*$, any feasible solution $\phi'$ of Eq. (24) satisfies:

$$\sum_{j=1}^{m} P_j \phi'_j + \sum_{j=1}^{m} Q_j (1 - \phi'_j) \geq 1 - c\beta^* = 1 - \|P - Q\|_{\mathrm{TV}}$$

with equality satisfied by the maximum likelihood rule $\phi^*$. Therefore, the min-budget optimality of the contract $t^*$ implies the power optimality of the hypothesis test $\phi^*$.

Conversely, let $\phi : \Omega \to [0,1]$ be an maximal-power hypothesis test. This provides a lower bound on the constraint in Eq. (24):

$$\sum_{j \in \Omega} P_j \phi_j + \sum_{j \in \Omega} Q_j (1 - \phi_j) \geq 1 - \|P - Q\|_{\mathrm{TV}}$$

By the Neyman-Pearson lemma, the bound is tight for the maximum likelihood rule $\phi^\star = \mathbb{1}\left[Q \geq P\right]$, and therefore the optimal objective $\beta^\star$ satisfies:

$$1 - c\beta^\star = 1 - \|P - Q\|_{\mathrm{TV}}$$

and hence $\beta^\star = \frac{\|P-Q\|_{\mathrm{TV}}}{c}$. Applying the transformation in Definition 3 yields the optimal contract $t_j^\star = \frac{c}{\|P-Q\|_{\mathrm{TV}}} \mathbb{1}\left[Q_j \geq P_j\right]$, showing that the corresponding contract is min-budget optimal if the corresponding hypothesis has optimal statistical power. $\square$

**Remark 1.** *Since the proof of Theorem 1 is independent of the Neyman-Pearson lemma, the argument proving the optimality of $\phi^*$ in the proof above implies the Neyman-Pearson lemma for finite $\mathcal{X}$.*

## B.5 More than two actions

**Definition 5** (All-or-nothing contract). *Let $B > 0$. An all-or-nothing contract $t : \Omega \to \mathbb{R}_{\geq 0}$ satisfies $t_j \in \{0, B\}$ for all $j \in \Omega$.*

### B.5.1 Binary outcomes

In this section, we show that every binary-outcome min-budget contract is all-or-nothing. This is a corollary of a more general lemma:

**Lemma 4.** *For any feasible min-budget contract $t^*$, there exists $j_0 \in \Omega$ such that $t^*_{j_0} = 0$.*

*Proof.* By contradiction, assume that $t^*_j > 0$ for all $j$, and denote $j_0 = \operatorname{argmin}_j t^*_j$. Denote $a = \min_j t_j$, and note that $a > 0$. Define the contract $\tilde{t}$ as follows:

$$\forall j : \tilde{t}_j = t_j - a$$

By definition, $\tilde{t}_j \geq 0$ for all $j$, and $\tilde{t}_{j_0} = 0$. Since $t^*$ is a feasible min-budget contract, it satisfies the min-budget LP in Eq. (6), and in particular the (IC) constraint:

$$\forall i \in [n-1] : \sum_j F_{i,j} t^*_j - c_i \leq \sum_j F_{n,j} t^*_j - c_n$$

Plugging in the definition $\tilde{t} = t^* - a$ and using the fact that $\sum_j F_{i,j} = 1$ for all $i \in [n]$, we obtain:

$$\forall i \in [n-1] : \sum_j F_{i,j} \tilde{t}_j - c_i \leq \sum_j F_{n,j} \tilde{t}_j - c_n$$

and therefore $\tilde{t}$ satisfies the (IC) constraint as well. This is a contradiction, since $\tilde{t}$ is a feasible contract with a lower required budget. From this we conclude that the optimal contract must satisfy $t_j = 0$ for some $j$. $\square$

**Corollary 1.** *In any min-budget contract design setting with two outcomes, the optimal contract is all-or-nothing.*

### B.5.2 Hardness: Basic definitions and construction

In this section, we show that finding optimal all-or-nothing contracts is NP-hard in the general case, by reduction from 3SAT.

**Definition 6** (Maximin design matrix). *For a contract design setting with action set $\mathcal{A}$, outcome space $\Omega$, target action $a^* \in \mathcal{A}$, and $c_a < c_{a^*}$ for all $a \in \mathcal{A} \setminus \{a^*\}$, the maximin design matrix $A$ is defined as:*

$$A_{a\omega} = \frac{F_{a^*,\omega} - F_{a,\omega}}{c_{a^*} - c_a} \tag{25}$$

**Claim 5.** *For a contract design problem, denote the maximin design matrix by $A$. An all-or-nothing min-budget contract $t^*$ can be written as $t^* = \phi^*/\beta^*$, where:*

$$\begin{aligned}
\phi^* &= \operatorname*{argmax}_{\phi \in \{0,1\}^{|\Omega|}} \min_{a \in \mathcal{A} \setminus \{a^*\}} (A\phi)_a \\
\beta^* &= \min_{a \in \mathcal{A} \setminus \{a^*\}} (A\phi^*)_a
\end{aligned} \tag{26}$$

*Proof.* By Lemma 2, Eq. (11), the min-budget contract design LP is equivalent to:

$$\max_{\phi \in [0,1]^{|\Omega|}, \beta \in \mathbb{R}_{\geq 0}} \beta$$

$$\text{s.t.}$$

$$\forall a \in \mathcal{A} \setminus \{a^*\} : \sum_{\omega \in \Omega} \frac{F_{a^*,\omega} - F_{a,\omega}}{c_{a^*} - c_a} \phi_\omega \geq \beta$$

Where $\beta = B^{-1}$. As $A_{a,\omega} = \frac{F_{a^*,\omega} - F_{a,\omega}}{c_{a^*} - c_a}$, it holds that $\sum_{\omega \in \Omega} \frac{F_{a^*,\omega} - F_{a,\omega}}{c_{a^*} - c_a} \phi_\omega = A\phi$. The LP above is therefore equivalent to:

$$\max_{\phi \in [0,1]^{|\Omega|}, \beta \in \mathbb{R}_{\geq 0}} \beta$$
$$\text{s.t.}$$
$$\forall a \in \mathcal{A} \setminus \{a^*\} : A_a \phi \geq \beta$$

as $\beta$ is a lower bound for every constraint, its optimal value is the maximal minimum attainable through variation of $\phi$:

$$\max_{\phi \in [0,1]^{|\Omega|}} \min_{a \in \mathcal{A} \setminus \{a^*\}} (A\phi)_a$$

and restricting the optimization space of $\phi$ to $\{0,1\}^{|\Omega|}$ yields the desired result. □

**Definition 7** (3-CNF – Conjunctive normal form). *A 3-CNF formula over $m$ variables and $n$ clauses is a boolean-valued function $\psi : \{0,1\}^m \to \{0,1\}$ of the form:*

$$\psi(x_1, \ldots, x_m) = \bigwedge_{i=1}^{n} (z_{i1} \vee z_{i2} \vee z_{i3})$$

*where $z_{ik} \in \{x_1, \ldots, x_m, \neg x_1, \ldots, \neg x_m\}$.*

We assume that variables in each clause are distinct. $\psi$ is satisfiable if and only if there exists $x \in \{0,1\}^m$ such that $\psi(x) = 1$.

**Definition 8** (Number of positives in clause $i$). *Given a 3-CNF $\psi$ and an assignment $x \in \{0,1\}^m$, denote by $\sigma_i(\psi, x) \in \{0, \ldots, 3\}$ the number of variables $z_{ik}$ in clause $i$ which evaluate to $1$ under assignment $x$.*

**Claim 6.** *A formula $\psi$ is satisfiable if and and only if there exists an assignment $x \in \{0,1\}^m$ such that $\min_{i \in [n]} \sigma_i(\psi, x) \geq 1$.*

*Proof.* If $\psi$ is satisfiable then there exists $x \in \{0,1\}^m$ such that $\psi(x) = 1$. Since $\psi$ is a 3-CNF, every clause $i$ must evaluate to 1, and therefore $\sigma_i(\psi, x) \geq 1$ for all $i \in [n]$. Conversely, if there exist an assignment $x$ such that $\min_{i \in [n]} \sigma_i(\psi, x) \geq 1$ then by definition $x$ satisfies every clause, and therefore the whole formula $\psi$. □

### B.5.3  Hardness: Reduction from 3SAT

Given a 3-CNF $\psi$ with $n$ clauses and $m$ variables, we define a min-budget contract design problem over actions $\mathcal{A} = [n+1]$ and outcome space $\Omega = [m] \cup \{\text{pos}, \text{neg}, \text{const}\}$. The target action is $a^* = n+1$, and the cost associated with action $i$ is:

$$c_i = \begin{cases} 1 & i = n+1 \\ 0 & \text{otherwise} \end{cases} \tag{27}$$

For simplicity of notations, let $\varepsilon > 0$, which will be assigned a suitable value later in the construction. The outcome distribution of the target action is denoted by $Q$, and defined as:

$$\forall j \in [m] : Q_j = \frac{\varepsilon}{m}$$
$$Q_{\text{pos}} = 1 - \varepsilon \left( 1 + \frac{3}{m} \right)$$
$$Q_{\text{neg}} = 0 \tag{28}$$
$$Q_{\text{const}} = \frac{3\varepsilon}{m}$$

Note that $Q$ is well-defined for $m \geq 3$ and $\varepsilon \leq \frac{1}{2}$.

For each $i \in [n]$, denote the number of negated variables in clause $i$ by $k_i \in \{0, \ldots, 3\}$. The distribution corresponding to the $i$-th action is denoted by $P^{(i)}$, and defined as:

$$\forall j \in [m] : P_j^{(i)} = \begin{cases} 0 & x_j \text{ exists in clause } i \\ \frac{2\varepsilon}{m} & \neg x_j \text{ exists in clause } i \\ \frac{\varepsilon}{m} & \text{otherwise} \end{cases}$$

$$P_{\text{pos}}^{(i)} = 0 \tag{29}$$

$$P_{\text{neg}}^{(i)} = 1 - \varepsilon \left( 1 + \frac{k_i}{m} \right)$$

$$P_{\text{const}}^{(i)} = (3 - k_i)\frac{\varepsilon}{m}$$

The distributions $P^{(i)}$ are well-defined when $m \geq 3$ and $\varepsilon \leq \frac{1}{2}$. For concreteness, set:

$$\varepsilon = \frac{1}{10} \tag{30}$$

and note that $Q_{\text{pos}}$ and $P_{\text{neg}}^{(i)}$ are both strictly positive for $m \geq 3$ and this value of $\varepsilon$.

Plugging equations (27, 28, 29) into Definition 6, the maximin design matrix $A^\psi \in \mathbb{R}^{n \times (m+3)}$ corresponding to the contract design problem above is given by:

$$\forall j \in [m] : A_{i,j}^\psi = \begin{cases} \frac{\varepsilon}{m} & x_j \text{ exists in clause } i \text{ and is not negated} \\ -\frac{\varepsilon}{m} & x_j \text{ exists in clause } i \text{ and is negated} \\ 0 & \text{otherwise} \end{cases}$$

$$A_{i,\text{pos}}^\psi = Q_{\text{pos}} \tag{31}$$

$$A_{i,\text{neg}}^\psi = -P_{\text{neg}}^{(i)}$$

$$A_{i,\text{const}}^\psi = k_i \frac{\varepsilon}{m}$$

**Definition 9** (Assignment normalized contract). *For an assignment $x \in \{0,1\}^m$, the corresponding normalized contract $\phi^x \in \{0,1\}^{m+3}$ is:*

$$\forall j \in [m] : \phi_j^x = x_j$$

$$\phi_{\text{pos}}^x = 1$$

$$\phi_{\text{neg}}^x = 0 \tag{32}$$

$$\phi_{\text{const}}^x = 1$$

**Claim 7.** *Let $\psi$ be a 3-CNF, and $x \in \{0,1\}^m$ an assignment. For all $i \in [n]$:*

$$\left( A^\psi \phi^x \right)_i = \frac{\varepsilon}{m} \sigma_i(\psi, x) + Q_{\text{pos}}$$

*Proof.* To prove this claim, plug $\phi_x$ from Definition 9 into $A^\psi$ defined in Eq. (31). To simplify notations, here we denote $A = A^\psi$, $\phi = \phi^x$, $z_i = \{z_{i1}, z_{i2}, z_{i3}\}$. We obtain:

$$(A\phi)_i = \sum_{\omega \in [m] \cup \{\text{pos}, \text{neg}, \text{const}\}} A_{i,\omega} \phi_\omega$$

$$= \underbrace{\sum_{j \in [m]} A_{i,j} \phi_j}_{= \frac{\varepsilon}{m}\left( \sum_{x_j \in z_i} \phi_j - \sum_{\neg x_{j'} \in z_i} \phi_{j'} \right)} + \underbrace{A_{i,\text{pos}}}_{=Q_{\text{pos}}} \underbrace{\phi_{\text{pos}}}_{=1} + \underbrace{A_{i,\text{neg}}}_{} \underbrace{\phi_{\text{neg}}}_{=0} + \underbrace{A_{i,\text{const}}}_{=k_i \frac{\varepsilon}{m}} \underbrace{\phi_{\text{const}}}_{=1}$$

$$= \frac{\varepsilon}{m} \left( \sum_{x_j \in z_i} \phi_j + \sum_{\neg x_{j'} \in z_i} (1 - \phi_{j'}) \right) + Q_{\text{pos}}$$

$$= \frac{\varepsilon}{m} \sigma_i(\psi, x) + Q_{\text{pos}}$$

$\square$

**Claim 8.** *For a given 3-CNF $\psi$, denote by $\phi^*$ the optimal solution for Eq. (26), with $A = A^\psi$. There exists an assignment $x^*$ such that $\phi^* = \phi^{x^*}$.*

*Proof.* For the maximin matrix $A^\psi$ defined in Eq. (31), the optimization objective in Eq. (26) is:

$$\min_{i \in [n]} \left( A^\psi \phi \right)_i$$

By the choice of $\varepsilon$ in Eq. (30), $A^\psi$ satisfies the following for all $i \in [n]$:

$$A_{i,\text{pos}} = Q_{\text{pos}} > 0$$
$$A_{i,\text{neg}} = -P_{\text{neg}}^{(i)} < 0$$
$$A_{i,\text{const}} = k_i \frac{\varepsilon}{m} > 0$$

Thus, any optimal solution $\phi^*$ must satisfy:

$$\phi_{\text{pos}}^* = 1$$
$$\phi_{\text{neg}}^* = 0$$
$$\phi_{\text{const}}^* = 1$$

Otherwise, the value of any $\left( A^\psi \phi \right)_i$ would strictly increase by changing the value of $\psi^*$ at the coordinates $\{\text{pos}, \text{neg}, \text{const}\}$ to the values specified above. This would increase the optimization objective $\min_{a \in \mathcal{A} \setminus \{a^*\}} \left( A^\psi \phi \right)_a$, in contradiction to the optimality of $\phi^*$.

Hence, denoting $x_j^* = \phi_j^*$ for all $j \in [m]$, we obtain $\phi^* = \phi^{x^*}$ as required. $\qquad\square$

**Claim 9.** *A 3-CNF $\psi$ is satisfiable if and only if the optimization problem in Eq. (26) with $A = A^\psi$ satisfies $\beta^* \geq Q_{\text{pos}} + \frac{\varepsilon}{m}$.*

*Proof.* If $\psi$ is satisfiable by assignment $x \in \{0,1\}^m$, then $\sigma_i(\psi, x) \geq 1$ for all $i \in [n]$ by Claim 6. Observe that $\beta^* = \left( A^\psi \phi^x \right)_{i^*}$ for some $i^* \in [n]$ by construction. Let $\phi^x$ denote the vector corresponding to the satisfying assignment, according to Definition 9. Apply Claim 7 to obtain:

$$
\begin{aligned}
\beta^* &= \left( A^\psi \phi^x \right)_{i^*} \\
&= \frac{\varepsilon}{m} \sigma_{i^*}(\psi, x) + Q_{\text{pos}} \\
&\geq Q_{\text{pos}} + \frac{\varepsilon}{m}
\end{aligned}
$$

Conversely, assume the optimal solution to Eq. (26) satisfies $\beta^* \geq Q_{\text{pos}} + \frac{\varepsilon}{m}$, and denote the vector attaining the optimal solution by $\phi^*$. By Claim 8, there exists an assignment $x^*$ such that $\phi^* = \phi^{x^*}$. Combining the lower bound on the value of the optimal solution with the result of Claim 7, we obtain that the following holds for all $i \in [n]$:

$$\left( A^\psi \phi^{x^*} \right)_i = \frac{\varepsilon}{m} \sigma_i(\psi, x^*) + Q_{\text{pos}} \geq \frac{\varepsilon}{m} + Q_{\text{pos}}$$

Therefore, it must hold that $\sigma_i(\psi, x^*) \geq 1$, and thus $x^*$ satisfies $\psi$ according to Claim 6. $\qquad\square$

### B.5.4   Proof of hardness

*Proof of Theorem 3.* By reduction from 3SAT. Given a 3-CNF $\psi$ with $n$ clauses and $m$ variables, construct in polynomial time the corresponding matrix $A^\psi$ as defined by Eq. (31), and apply an all-or-nothing min-budget contract solver according to Eq. (26) to obtain the optimal solution. The validity of the LP is given by Eq. (26). By Claim 9, the formula $\psi$ is satisfiable if and only if the value attained in the optimization is at least $\frac{\varepsilon}{m} + Q_{\text{pos}}$, where $m$ is the number of variables in $x$, and $\varepsilon, Q_{\text{pos}}$ are constants defined in Eq. (30), Eq. (28) respectively. $\qquad\square$

## B.6  The single binding action algorithm

To prove the soundness of SBA, we first make the following definition:

**Definition 10** (($i'$-IC) relaxation). *Consider a MIN-BUDGET LP with target action $i \in \mathcal{A}$, as given by Eq. (6). For any action $i' \neq i$, the ($i'$-IC) relaxation of the min-budget problem is given by eliminating all (IC) constraints except for the one corresponding to action $i'$.*

*Proof of Proposition 2.* Denote the target action by $i \in \mathcal{A}$. On each iteration of the loop, the algorithm considers some action $i' \neq i$, and applies Theorem 1 on the ($i'$-IC) relaxation of the original MIN-BUDGET LP. Since the relaxed LP has only one (IC) constraint, its optimal solution, denoted by $t^*(i', i)$, is given by Eq. (5).

By Definition 10, the feasible region of the original LP lies within feasible region corresponding to its ($i'$-IC) relaxation. Thus, if an optimal solution of the relaxed LP lies within the feasible region of the original LP, then it is also a global optimum of the original LP. $t^*(i', i)$ lies within the feasible region of the original LP is it satisfies the remaining (IC) constraints—a condition equivalent to the notation $a(t^*(i', i)) = i$ by Eq. (2). The SBA algorithm terminates successfully only in such cases, and therefore it is sound. $\qquad\square$

**A note on ties in SBA.** Denote the target action by $i$. To simplify presentation, the algorithm presentation in Section 3.4 implicitly assumes that required budgets $\|t^*(i', i)\|_\infty$ are distinct for every pair of actions $(i', i)$, and therefore the return value of SBA is well-defined. In case of ties, the return value is not well-defined (as the iteration order is not well-defined), but small a modification to the algorithm allows ties to be broken explicitly without affecting other properties of the algorithm: In case of ties in optimal required budget, the soundness of the algorithm and the all-of-nothing property of the returned contracts is not affected. However, the exact functional form of the returned contract may be affected by the iteration order. In case such issue becomes relevant, the SBA algorithm can be modified to first collect a set of optimal contracts (instead of immediately returning when an optimal feasible contract is found), and then select one of the optimal contracts based on the desired criteria. As the proof of Proposition 2 does not depend on iteration order, soundness will not be affected, and worst-case time complexity will remain the same.

## B.7  MLRP

In this section, we explore contract design under the MLRP assumption, and prove Theorem 4. We first recall the statistical notion of monotone likelihood ratio:

**Definition 11** (Mononote Likelihood Ratio – MLR). *The distributions $P, Q \in \Delta(\Omega)$ have Monotone Likelihood Ratio when $\frac{Q_j}{P_j}$ is monotonically increasing for all $j \in \Omega = \{0, \ldots, m\}$. We denote this by $P \prec Q$.*

An introduction to MLR and its relation to statistical hypothesis testing is provided in Lehmann et al. [40, Section 3.4]. Using this notation, we can reformulate Definition 1:

**Definition 12** (MLRP; reformulation of Def. 1 using MLR notation). *A principal-agent problem satisfies the Monotone Likelihood Ratio Property if $F_{i'} \prec F_i$ for every pair of actions $i, i'$ such that $c_{i'} < c_i$.*

**Claim 10.** *Let $P, Q \in \Delta([m])$ such that $P \prec Q$. Then $P_0 > Q_0$ and $P_m < Q_m$.*

*Proof.* For the first inequality, assume by contradiction that $Q_0 \geq P_0$. Combining with the MLR property, this assumption implies that $Q_j \geq P_j$ for all $j \in \{0, \ldots, m\}$. As the outcome distributions $P, Q$ are distinct, there exists at least one $j$ for which $Q_j > P_j$. Summing over $j$ yields: $\sum_{j \in \Omega} Q_j > \sum_{j \in \Omega} P_j$, which is a contradiction, as the inequality is strict while both sides sum to 1. The proof for the second inequality follows in the same way. $\qquad\square$

**Definition 13** (MLR crossing point $j^*_{P,Q}$). *Let $P, Q \in \Delta(\Omega)$ such that $P \prec Q$. The crossing point of $Q$ over $P$ is the outcome $j^* \in \{0, \ldots, m\}$ such that $Q_{j^*-1} < P_{j^*-1}$ and $Q_{j^*} \geq P_{j^*}$. When distribution are not clear from context, we denote $j^* = j^*_{P,Q}$*

In words, the crossing point $j^*$ is the outcome such that for every lower outcome, $P$ is strictly more likely than $Q$, and starting with this outcome $Q$ is weakly more likely. By claim 10, the MLR crossing point $j^*$ is uniquely defined for any $P \prec Q$, and it holds that $j^* > 0$.

**Definition 14** (Survival function). *Given $P \in \Delta([m])$, the survival function $\mathbb{S}_P(\cdot) : \Omega \to [0,1]$ is defined as:*

$$\mathbb{S}_P(j) = \mathbb{P}_{j' \sim P}[j' > j] = \sum_{j'=j+1}^{m} P_i$$

Informally, the survival function is 1 minus the distribution's CDF. This function is useful in our context, since the total variation distance between two distributions (as appearing in Theorem 5) can be represented as the difference between their survival functions when they satisfy MLR. The intuitive reason for this is that MLR means the distributions are "single-crossing".

**Claim 11** (Total variation under MLR). *Let $P, Q \in \Delta(\Omega)$ such that $P \prec Q$. It holds:*

$$\|P - Q\|_{\mathrm{TV}} = \mathbb{S}_Q(j^*_{P,Q} - 1) - \mathbb{S}_P(j^*_{P,Q} - 1)$$

*Proof.* By Claim 4, it holds that:

$$\|P - Q\|_{\mathrm{TV}} = \sum_{j \in \{0,\dots,m\}} (Q_j - P_j)^+$$

Using the $j^*$ notation (see Definition 13), we obtain:

$$\sum_{j \in \{0,\dots,m\}} (Q_j - P_j)^+ = \sum_{j=j^*_{P,Q}}^{m} Q_j - P_j$$

And the result is obtained by applying the definition of survival function (see Definition 14). $\qquad\square$

### B.7.1 Two-action contracts with MLRP

Combining Theorem 5 with Claim 10 leads to a characterization of threshold contracts in the case of two actions:

**Claim 12.** *For any two-action contract design problem satisfying MLRP, the optimal contract is a threshold contract:*

$$t^*_j = \frac{c}{\|F_2 - F_1\|_{\mathrm{TV}}} \mathbb{1}\left[j \geq j^*_{F_1,F_2}\right]$$

*where $j^*_{F_1,F_2} = \min\{j \in \{0,\dots,m\} \mid F_{2,j} \geq F_{1,j}\}$ as in Definition 13.*

*Proof.* Combining the result of Claim 10 with the monontonicity assumption, the likelihood ratio $F_{2,j}/F_{1,j}$ crosses 1 exactly once. Denote the crossing point by $j^*$, and apply Theorem 5 to obtain the optimal contract. $\qquad\square$

### B.7.2 Two-outcome contracts with MLRP

When there are more than two actions, assume without loss of generality that the contract aims to implement the last action $n$. The following claim establishes the existence of theshold contracts in the two-outcome setting $|\Omega| = 2$. In contrast to corollary 1, the proof in constructive, and yields a concrete contract:

**Claim 13.** *For any contract design problem satisfying MLRP with $n > 2$ actions and $m = 2$ outcomes, the optimal min-budget contract is a threshold contract.*

*Proof.* By Claim 2, the dual LP for two outcomes is given by:

$$\max_{\lambda \in \mathbb{R}^{n-1}_{\geq 0}, \mu \in \mathbb{R}^2_{\geq 0}} \sum_{i=1}^{n-1} (c_n - c_i)\lambda_i$$

s.t.

$$\forall j \in \{1, 2\} : \sum_{i=1}^{n-1} (F_{n,j} - F_{i,j}) \lambda_i \leq \mu_j \tag{33}$$

$$\sum_{j \in \{1,2\}} \mu_j \leq 1$$

From Claim 10, we obtain that $F_{n,1} - F_{i,1} < 0$ and $F_{n,2} - F_{i,2} > 0$ for all $i \in [n-1]$, and therefore the first constraint in Eq. (33) is always satisfied for $j = 1$. Simplifying Eq. (33) we obtain:

$$\max_{\lambda \in \mathbb{R}^{n-1}_{\geq 0}, \mu \in \mathbb{R}_{\geq 0}} \sum_{i=1}^{n-1} (c_n - c_i)\lambda_i$$

s.t. $\tag{34}$

$$\sum_{i=1}^{n-1} (F_{n,2} - F_{i,2}) \lambda_i \leq 1$$

which is maximized by allocating all budget to the $\lambda_i$ maximizing the "bang for buck". The dual objective is therefore given by:

$$
\begin{aligned}
B^* &= \max_{\lambda \in \mathbb{R}^{n-1}_{\geq 0}, \mu \in \mathbb{R}_{\geq 0}} \sum_{i \in [n-1]} (c_n - c_i)\lambda_i \\
&= \max_{i \in [n-1]} \frac{c_n - c_i}{F_{n,2} - F_{i,2}}
\end{aligned}
\tag{35}
$$

To see that a threshold contract is optimal, let $t_j^* = B^* \cdot \mathbb{1}[j = 1]$. Primal objective is $B^*$, and the contract is feasible if the primal LP is feasible. The (IC) constraint in Eq. (6) can be written as:

$$\forall i \in [n-1] : \frac{\sum_{j=1}^2 (F_{n,j} - F_{i,j}) t_j}{c_n - c_i} \geq 1$$

and indeed for every action $i \in [n-1]$:

$$
\begin{aligned}
\frac{\sum_{j=1}^2 (F_{n,j} - F_{i,j}) t_j}{c_n - c_i} &= \frac{F_{n,2} - F_{i,2}}{c_n - c_i} B^* \\
&= \frac{F_{n,2} - F_{i,2}}{c_n - c_i} \max_{i \in [n-1]} \frac{c_n - c_i}{F_{n,2} - F_{i,2}} \\
&\geq 1
\end{aligned}
$$

And therefore $t_j^*$ is feasible in the primal problem, and also optimal by strong LP duality. Also note that the resulting contract coincides exactly with the optimal min-pay contract as attained by Dütting et al. [19, Lemma 7]. $\qquad \square$

### B.7.3 General contracts with MLRP

In this section, we explore min-budget contracts in MLRP settings with more than two actions and more than two outcomes. We start with a negative example, showing that the MLRP assumption is not sufficient for establishing the optimality of threshold contracts:

**Claim 14.** *For $|\Omega| > 2$, there exists a design problem satisfying MLRP for which the optimal contract is not a threshold.*

*Proof.* Consider the following contract design problem:

$$
\begin{aligned}
F_1 &\sim \text{Binomial}(10, 0.5) & c_1 &= 0 \\
F_2 &\sim \text{Binomial}(10, 0.65) & c_2 &= 0.45 \\
F_3 &\sim \text{Binomial}(10, 0.8) & c_3 &= 1
\end{aligned}
$$

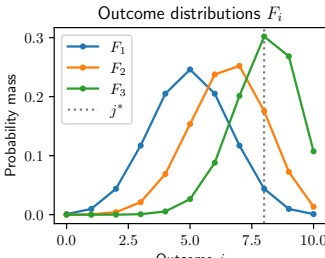 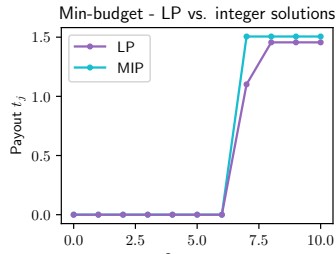 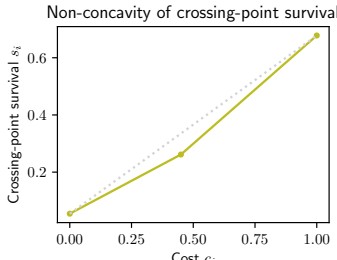

Figure 7: Graphical illustration of Claim 14. **(Left)** Outcome distributions $F_1, F_2, F_3$ and MLR crossing point $j^*_{F_2, F_3}$. **(Center)** The min-budget contract $t^*_j$, given by numerically solving Eq. (11) (purple), and numerically solving Eq. (11) while imposing integer constraints $\phi_j \in \{0, 1\}$ (cyan). Note that the fractional LP solution achieves lower max payout. **(Right)** Crossing-point survival $s_i = \mathbb{S}_{F_i}(j^*_{F_i, F_n} - 1)$ as a function of cost $c_i$ (see Definition 15). Note that the function is not concave, thus the sufficient condition given in Theorem 4 does not hold in this case.

The distributions are binomial, and therefore the contract satisfies MLRP. Numerically solving Eq. (11) yields the following fractional contract:

$$t^*_{\mathrm{LP}} = (0, 0, 0, 0, 0, 0, 0, 1.1, 1.46, 1.46, 1.46)$$

Numerically solving Eq. (11) while imposing integer constraints $\phi_j \in \{0, 1\}$ yields the following contract, which has higher max payout:

$$t^*_{\mathrm{IP}} = (0, 0, 0, 0, 0, 0, 0, 1.51, 1.51, 1.51, 1.51)$$

Thus for this case any threshold contract is min-budget suboptimal. A graphical illustration of the proof is provided in Fig. 7. $\qquad\qquad\square$

**Remark 2.** *The proof of Claim 14 is provided with 10 outcomes ($|\Omega| = 10$) for ease of graphical interpretation, however a minimal counterexample may also be constructed using only 3 outcomes. For example, consider the following design problem:*

$$
\begin{aligned}
F_1 &= (0.5, 0.3, 0.2) & c_1 &= 0 \\
F_2 &= (0.3, 0.4, 0.3) & c_2 &= 0.45 \\
F_3 &= (0.1, 0.35, 0.55) & c_3 &= 1
\end{aligned}
$$

*The proof for this case follows in the same way, where numerical computation yields the contracts:*

$$t^*_{\mathrm{LP}} = (0, 1.9, 2.6)$$
$$t^*_{\mathrm{IP}} = (0, 2.7, 2.7)$$

**Remark 3.** *We also note that a counterexample with 2 outcomes is not possible due to Claim 13.*

The negative example shows that even with MLRP, the optimal contract incentivizing the highest implementable action needs to rule out deviations of the agent to all other actions, and not just to the second-highest one. This makes the contract complex. We identify a natural economic condition that is sufficient for considering only a single deviation (to the second-highest action), resulting in a simple contract. Note that considering a single such deviation is equivalent to restricting the action space to actions $\{n - 1, n\}$.

### B.8 Sufficiency condition for more than two actions

For the following definition, recall the definition of MLR crossing point $j^*$ (Def. 13), and survival function $\mathbb{S}$ (Def. 14):

**Definition 15** (Concave-MLRP; Formal statement of Definition 2). *For a contract design setting satisfying MLRP, denote by $j^* = j^*_{F_n, F_{n-1}}$ the crossing point of the outcome distribution corresponding to the highest action $F_n$, and the outcome distribution corresponding to the second-highest action $F_{n-1}$. For any action $i \in [n]$, the* crossing-point survival $s_i$ *is defined as:*

$$s_i = \mathbb{S}_{F_i}(j^* - 1)$$

In words, $s_i$ is the probability to get an outcome at or above crossing-point $j^*$ according to outcome distribution $F_i$. From an economic perspective, for every action $i \in [n]$, $s_i$ is the agent's probability to receive any nonzero payment from choosing action $i$, if the contract is designed by restricting the action-space to actions $\{n-1, n\}$. Note however that the definition of $s_i$ only depends on the outcome distribution $F_{ij}$, and does not require solving the contract design problem.

### B.8.1 Binomial power-law curves satisfy Concave-MLRP

Recent theoretical work on learning curves has focused mainly on power-law expected learning curves of the form $a_n = 1 - \beta n^{-\gamma}$ [11, 33]. In addition, these curves were also found to provide a good fit for certain practical applications [49, POW2]. In this section, we show that a stochastic generalization of these curves satisfies the Concave-MLRP property:

**Definition 16** (Power-law stochastic learning curve). *Let $\beta, \gamma \in \mathbb{R}_{\geq 0}$, and $m \in \mathbb{N}$. A delegated learning setting with actions $\mathcal{A}$ has a realizable power-law stochastic learning curve with parameters $\beta, \gamma$ if $F_i = \text{Binomial}\left(1 - \beta n_i^{-\gamma}, m\right)$ for all $n_i \in \mathcal{A}$.*

For the proof, we also recall the definition of the regularized incomplete beta function:

**Definition 17** (Regularized incomplete beta function). *For $k_1, k_2 \geq 1$, the regularized incomplete beta function is defined as:*

$$I_p(k_1, k_2) = \frac{\int_0^p t^{k_1-1}(1-t)^{k_2-1}\mathrm{d}t}{\int_0^1 t^{k_1-1}(1-t)^{k_2-1}\mathrm{d}t}$$

We also recall that $I_p$ is related to the survival function of the binomial distribution [e.g., 1, 6.6.4]:

$$I_p(k+1, n-k) = \mathbb{P}_{x \sim \text{Binomial}(p,n)}[x > k]$$

**Claim 15.** *Let $\beta, \gamma \in \mathbb{R}_{\geq 0}$, $m \in \mathbb{N}$. A delegated learning setting with a power-law stochastic learning curve $F_i = \text{Binomial}(1 - \beta n_i^{-\gamma}, m)$, action costs $c_i = n_i$, and $\min_i n_i \geq \left(\beta(m+\gamma^{-1})/1+\gamma^{-1}\right)^{\frac{1}{\gamma}}$ satisfies the Concave-MLRP assumption.*

*Proof.* Denote the expected accuracy of each action by $a_n = 1 - \beta n^{-\gamma}$. The survival function of the binomial distribution is given by:

$$s_n = I_{a_n}\left(j^*, m + 1 - j^*\right)$$

Where $I_{a_n}$ is the regularized incomplete beta function (Definition 17). With slight abuse of notation, we treat $a_n$ and $s_n$ as continuous functions of $n$. Taking the derivative by $n$ and ignoring the constant denominator, we obtain:

$$\frac{\mathrm{d}s_n}{\mathrm{d}n} \propto a_n^{j^*-1}(1-a_n)^{m-j^*}\frac{\mathrm{d}a_n}{\mathrm{d}n}$$

Plugging the functional form of $a_n$:

$$\frac{\mathrm{d}s_n}{\mathrm{d}n} \propto \left(1 - \beta n^{-\gamma}\right)^{j^*-1}\left(\beta n^{-\gamma}\right)^{m-j^*}\beta\gamma n^{-(\gamma+1)}$$

$$\propto \left(\beta n^{-\gamma}\right)^{m-j^*+1+\gamma^{-1}}\left(1 - \beta n^{-\gamma}\right)^{j^*-1}$$

As a function of $\beta n^{-\gamma}$, the function $\frac{\mathrm{d}s_n}{\mathrm{d}n}$ attains its extremum at:

$$\beta\tilde{n}^{-\gamma} = \frac{m+1-j^*+\gamma^{-1}}{m+\gamma^{-1}}$$

$$\tilde{n} = \left(\frac{\beta(m+\gamma^{-1})}{m+1-j^*+\gamma^{-1}}\right)^{\frac{1}{\gamma}}$$

And therefore the function $s_n$ has an inflection point at $\tilde{n}$ (see Fig. 8). From the upper bound $j^* \leq m$ we obtain:

$$\tilde{n} \leq \left(\frac{\beta(m+\gamma^{-1})}{1+\gamma^{-1}}\right)^{\frac{1}{\gamma}} = n_0$$

And as $\min_i n_i \geq n_0$, the function $s_n$ is convex as a function of $c_n$ for all $n \in \mathcal{A}$. $\square$

The proof is illustrated in Fig. 8.

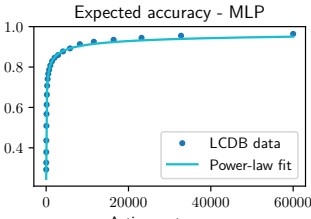 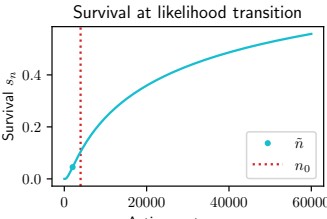 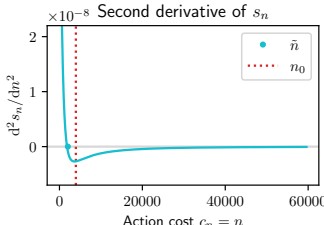

Figure 8: Graphical intuition for the sufficiency condition in Claim 15. **(Left)** Expected accuracy curve for MNIST-784 MLP. Blue dots represent empirical data from the LCDB dataset, cyan curve represents power-law fit ($a_n = 1 - \beta n^{-\gamma}$, with $\hat{\beta} = 1.89$, $\hat{\gamma} = 0.33$). **(Center)** Crossing point survival $s_i$ as a function of cost $c_i$ (see Definition 15), for $m = 30$. Cyan dot represents inflection point $\tilde{n} \approx 2022$. Red vertical line represents the inflection point bound $n_0 \approx 3957$ suggested by the proof. It can be observed that the curve is concave for all $n > \tilde{n}$. **(Right)** Second derivative of the survival function $s_n$, illustrating the position of the bound $n_0$ in relation to the inflection point $\tilde{n}$.

### B.8.2 Proof of sufficiency theorem

**Theorem 6** (Concave-MLRP implies threshold contracts; formal statement of Theorem 4). *For a contract design problem satisfying MLRP, consider $s_i$ as a function of cost $c_i$. If this function is concave, then the optimal contract is a threshold contract. Furthermore, the contract is successfully recovered by the SBA algorithm.*

*Proof.* To prove this claim, we construct a relaxed version of the min-budget LP (Eq. (6)), and apply Theorem 5 in order to solve it. We then show that this solution is also feasible for the original LP.

Given the min-budget LP, construct a relaxed LP by eliminating (IC) constraints for all $i \leq n - 2$:

$$\min_{t \in \mathbb{R}_{\geq 0}^{|\Omega|}, B \in \mathbb{R}_{\geq 0}} B$$

$$\text{s.t.}$$

$$\forall j \in \Omega : t_j \leq B \tag{36}$$

$$\sum_{j \in \Omega} F_{n-1,j} t_j - c_{n-1} \leq \sum_{j \in \Omega} F_{n,j} t_j - c_n$$

Eq. (36) only depends on $F_n$ and $F_{n-1}$, and therefore claim 12 can be applied to obtain the optimal min-budget contract:

$$t_j^* = \frac{c_n - c_{n-1}}{\|F_n - F_{n-1}\|_{\text{TV}}} \mathbb{1}\left[F_{n,j} \geq F_{n-1,j}\right] \tag{37}$$

Let $j^* = \min\left\{j \in \{0, \ldots, m\} \mid F_{n,j} \geq F_{n-1,j}\right\}$ as in Definition 13, and denote $s_i = \mathbb{S}_{F_i}(j^* - 1)$. By claim 11, we obtain:

$$\|F_n - F_{n-1}\|_{\text{TV}} = s_n - s_{n-1}$$

For all $i < n$, denote:

$$b_i = \frac{c_n - c_i}{s_n - s_i}$$

Using the definition of $b_i$, Eq. (37) can be rewritten as:

$$t_j^* = b_{n-1} \mathbb{1}\left[j \geq j^*\right]$$

By assumption, $s_i$ is a concave function of $c_i$, and therefore the function $b_i$ is monotonically non-decreasing for all $i < n$:

$$b_{n-1} \geq b_i$$

Dividing both sides by $b_i$, we obtain:

$$\frac{b_{n-1}}{b_i} \geq 1$$

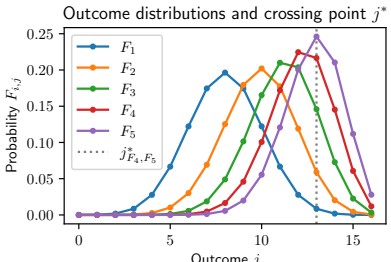 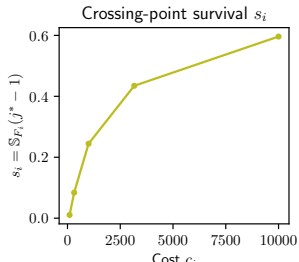 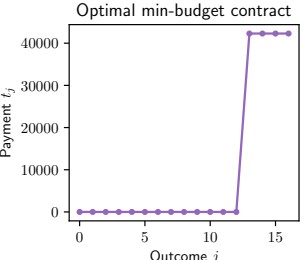

Figure 9: Graphical intuition for the sufficiency condition in Theorem 4 (restated in Theorem 6). **(Left)** Outcome distribution in a contract design setting satisfying MLRP, generated by a series of binomial distributions with power-law expectation $\text{acc}_D(h_n) = 0.9 - 0.4\left(\frac{n}{100}\right)^{-0.3}$. The gray dotted line represents the MLR crossing point $j^*_{F_4,F_5}$ (see Definition 13). **(Center)** Crossing point survival $s_i$ as a function of cost $c_i$ (see Definition 15). It can be observed that the curve is concave. **(Right)** The min-budget contract implementing action 5. It is a threshold contract, as guaranteed by Theorem 4. Compare to Fig. 7, where the sufficiency condition does not hold.

Plugging in the definition of $b_i$, and multiplying both sides by $(c_n - c_i)$:

$$b_{n-1}(s_n - s_i) \geq c_n - c_i \tag{38}$$

By definition of $t_j^*$, the expected payout of contract $t_j^*$ under action $i' \in [n]$ is given by:

$$\sum_{j=0}^{m} t_j^* F_{i',j} = b_{n-1} \sum_{j=j^*}^{m} F_{i',j} = b_{n-1} s_{i'} \tag{39}$$

Plugging Eq. (38) into Eq. (39) yields:

$$\sum_{j=0}^{m} t_j F_{n,j} - \sum_{j=0}^{m} t_j F_{i,j} \geq c_n - c_i \tag{40}$$

As Eq. (40) is identical to the (IC) constraint in Eq. (6), we obtain that $t_j^*$ satisfies all the (IC) constraints in the original LP. From this we conclude that $t_j^*$, which is a threshold contract, is also an optimal contract with the respect to the full min-budget LP.

For the second part of the claim, note that the SBA algorithm also attempts to solve Eq. (36) on the iteration corresponding to action $N - 1$. As the solution of Eq. (36) is guaranteed to be feasible for the original LP under Concave-MLRP, the SBA successfully recovers it. $\qquad\square$

### B.9 Min-budget contracts with guaranteed minimum payout

Our problem setting assumes that the agent selects its action rationally, as described in Eq. (2). However, in some practical settings the agent may be risk averse to some extent, and require guaranteed minimum payment from the contract. In this section, we show that the rationality assumption is made without loss of generality in such cases: Lemma 5, which we prove below, shows that any optimal contract with guaranteed minimum payout can be represented as the sum of a min-budget contract without guaranteed payout, and a constant representing the payout guarantee.

**Definition 18** (Guaranteed minimum payout). *Let $\delta \geq 0$. A contract $t \in \mathbb{R}_{\geq 0}^{|\Omega|}$ has guaranteed payout of size $\delta$ if $t_j \geq \delta$ for all $j \in \Omega$.*

**Lemma 5.** *A contract $t$ is a min-budget contract with guaranteed payout $\delta$ requiring budget $B$ if any only if there exist a min-budget contract $t'$ (without guaranteed payout) requiring budget $B'$ such that $t = t' + \delta$ and $B = B' + \delta$.*

*Proof.* When agents require a guaranteed minimum payout $\delta \geq 0$, we add a constraint to the MIN-BUDGET linear program defined in Eq. (6), such that an optimal min-budget contract $t$ with

guaranteed payout $\delta$ is an optimal solution to the following linear program:

$$\min_{t\in\mathbb{R}^{|\Omega|}_{\geq 0},B\in\mathbb{R}_{\geq 0}} B$$

s.t.

$$\forall j \in \Omega : t_j \leq B \qquad\qquad\text{(BUDGET)}$$

$$\forall i' \neq i : \sum_{j\in\Omega} F_{i',j}t_j - c_{i'} \leq \sum_{j\in\Omega} F_{i,j}t_j - c_i \qquad\qquad\text{(IC)}$$

$$\forall j \in \Omega : t_j \geq \delta \qquad\qquad\text{(MINWAGE)}$$

(41)

To prove the first direction of the equivalence, assume that $t$ is a min-budget contract with guaranteed payout $\delta$ and budget $B$, and thus an optimal solution of Eq. (41). Define the variable transformation:

$$t' = t - \delta$$
$$B' = B - t_0$$

(42)

By Eq. (42), the (BUDGET) constraint in Eq. (41) transforms into:

$$\forall j \in \Omega : t_j \leq B$$
$$\Leftrightarrow t'_j + \delta \leq B' + \delta$$
$$\Leftrightarrow t'_j \leq B'$$

(43)

The (IC) constraint transforms into:

$$\forall i' \neq i : \sum_{j\in\Omega} F_{i',j}t_j - c_{i'} \leq \sum_{j\in\Omega} F_{i,j}t_j - c_i$$
$$\Leftrightarrow \sum_{j\in\Omega} F_{i',j}(t'_j + \delta) - c_{i'} \leq \sum_{j\in\Omega} F_{i,j}(t'_j + \delta) - c_i$$
$$\Leftrightarrow \sum_{j\in\Omega} F_{i',j}t'_j - c_{i'} + \delta\underbrace{\sum_{j\in\Omega} F_{i',j}}_{=1} \leq \sum_{j\in\Omega} F_{i,j}t'_j - c_i + \delta\underbrace{\sum_{j\in\Omega} F_{i,j}}_{=1}$$
$$\Leftrightarrow \sum_{j\in\Omega} F_{i',j}t'_j - c_{i'} \leq \sum_{j\in\Omega} F_{i,j}t'_j - c_i$$

(44)

And the (MINWAGE) constraint transforms into:

$$\forall j \in \Omega : t_j \geq \delta$$
$$\Leftrightarrow t'_j \geq 0$$

(45)

Plugging back the transformed constraints (44, 43, 45) into Eq. (41), we obtain that $(t', B')$ is an optimal solution of the original MIN-BUDGET linear program Eq. (6), and therefore $t'$ is an optimal min-budget contract without minimum guaranteed payout.

Conversely, assume that $t'$ is an optimal min-budget contract (without minimum guaranteed payout), satisfying the MIN-BUDGET linear program in Eq. (17) with budget $B'$. Apply the inverse transformation $t = t' + \delta$ and $B = B' + \delta$, and the equivalence relations (44, 43, 45) in the inverse direction to obtain that $t, B$ is an optimal solution to Eq. (41). $\qquad\square$

## C Experimental details

### C.1 Data

**LCDB.** We base our main experimental environment on the LCDB dataset [43], which includes a large collection of stochastic learning curves for multiple learning algorithms and classification benchmark datasets. For each learning method and benchmark dataset, the database includes held-out accuracy measurements, obtained for exponentially-increasing training set sizes $n \in \{2^4 = 16, \ldots, \text{round}(2^{k/2}), \ldots, 2^{15} = 32,768\}$, with multiple repetitions per $n$ obtained by cross-validation. For all learning curves we consider in our analysis, and for each $n$, the number of

repetitions provided in the dataset is in the range $R \in \{5, \ldots, 125\}$, where the specific number of repetitions in LCDB depends on the algorithm and benchmark dataset (see the dataset documentation for additional details). For each trained classifier, each accuracy point on the learning curve is estimated using 5,000 held-out samples. Formally, and using our notation, for each learning algorithm Alg (e.g. MLP), dataset $D$ (e.g. MNIST), and training set size $n$, LCDB provides a set of accuracy estimates $\{a_n^{1,\text{Alg},D}, \ldots, a_n^{R,\text{Alg},D}\}$, such that each $a_n$ is distributed according to $a_n \sim \text{acc}_D(h_n)$, where $h_n$ is a classifier trained using training set $S \sim D^n$ and learning algorithm Alg (see Section 2 for definition of $\text{acc}_D(h_n)$).

**Benchmark dataset and algorithms.** In our main analysis, we focus on the popular MNIST dataset [39, OpenML 554], and on multilayer perceptrons (MLP) and gradient-boosted decision trees (GBDT) as representative classifiers. For all classifiers, results are obtained for the respective default `scikit-learn` [44] implementations (e.g. `MLPClassifier` and `GradientBoostingClassifier` for MLP and GBDT, respectively).

**Delegated learning settings from empirical data.** For a given validation set size $m$ where $m = |V|$, we instantiate a contract design task $(\mathcal{A}, c, \Omega, F)$ as follows: (i) the set of actions with the set of training set sizes provided by LCDB ($\mathcal{A} = \{2^4, 2^{4.5}, \ldots, 2^{15}\}$); (ii) action costs are set to fixed per-unit cost, i.e., $c_n = n$; and (iii) the distribution $F$ over outcomes $\Omega$ is associated with a binomial mixture distribtuion, resulting from applying bootstrap sampling to empirical error measurements:

$$f_n = \frac{1}{R} \sum_{r=1}^{R} \text{Binomial}(m, a_n^{r,\text{Alg},D})$$

where $a_n$ are the accuracy estimates defined above. Fig. 2 (Left) shows an example of such a setting, obtained by applying the above procedure to data describing learning curves for the MLP algorithm trained on the MNIST benchmark dataset.

## C.2 Implementation details

**Code.** We implement our code in Python. Our code relies on `Pyomo` [13, 27] and GLPK for solving linear and mixed-integer programs.
Code is available at: https://github.com/edensaig/delegated-classification.

**Hardware.** All experiments were run on a single laptop, with 16GB of RAM, M1 Pro processor, and with no GPU support.

**Runtime.** A single run consisting the entire pipeline takes roughly 5 minutes. The main bottleneck is running the LP solvers, taking roughly 70% of runtime to compute.

### C.2.1 Contract design solvers

To find the optimal contracts, we implement and compare several different solvers:

- **LP solver**: To find min-budget contracts given outcome distributions $\{f_n\}$ and costs $\{c_n\}$, we solve the MIN-BUDGET LP (Eq. (6)) directly using `Pyomo` and GLPK. Given budget $B$, budget-optimal contracts are obtained by iterating through the actions set $\mathcal{A}$, invoking the min-budget solver on each target action $n \in \mathcal{A}$, enforcing incentive compatibility against all actions $n'$ which satisfy $\alpha_{n'} < \alpha_n$, and taking the action yielding the maximal expected accuracy within budget (see Appendix B.1). In our code, the LP solver is implemented within the `MinBudgetContract` class. Typical running time for a single problem instance: $10^{-1}$s.

- **SBA solver**: Implementation of the single binding action algorithm presented in Section 3.4. The local solver is implemented within the `MinBudgetSingleBindingConstraintContract` class. Typical running time for a single problem instance: $10^{-4}$s.

- **Hybrid solver**: A meta-solver implementing the 'try SBA first' computational approach. The solver starts by invoking the SBA solver, and applies the LP solver if the former fails. In our code, the LP solver is implemented within the `MinBudgetHybridContract` class. Running time depends on whether SBA is applicable.

- **MIP solver**: To find optimal all-or-nothing contracts, we solve the MIN-BUDGET LP in its statistical formulation (Eq. (11)) while restricting $\phi_j \in \{0, 1\}$. This turns the LP into a mixed-integer program, which can also be solved using GLPK. In our code, the IP solver is implemented within the `MinBudgetStatisticalContract` class. Typical running time for a single problem instance: $10^{-1}$s.

- **Min-pay LP solver**: To compare between min-budget and min-pay contracts (Appendix B.3), min-pay contracts were obtained by solving the MIN-PAY LP (Eq. (17)) using `Pyomo` and GLPK, similar to the full min-budget LP solver. In our code, the min-pay is implemented within the `MinPayContract` class. Running time is similar to the other LP-based methods.

- **Full enumeration solver**: To find all-or-nothing contracts for low-dimensional problems and avoid numerical instabilities, we implement a solver which performs full enumeration of all-or-nothing contracts. By the statistical variable transformation in Definition 3, the (IC) constraint in the MIN-BUDGET LP translates to $\sum_{j \in \Omega} \left( F_{i,j} - F_{i',j} \right) \phi_j \leq (c_i - c_{i'})\beta$, where $i$ is the target action, $i' \in \mathcal{A} \setminus \{i\}$, and the objective is minimize $\beta$ (see Eq. (11)). Thus, when $\phi \in \{0, 1\}^\Omega$ is fixed, the optimal $\beta$ is given by:

$$\beta^*(\phi) = \min_{i' \in \mathcal{A} \setminus \{i\}} \frac{\sum_{j \in \Omega} \left( F_{i,j} - F_{i',j} \right) \phi_j}{c_i - c_{i'}}$$

and optimizing $\beta^*$ over $\phi \in \{0, 1\}^\Omega$ yields an optimal all-or-nothing contract. The full enumaration solver is implemented within the `FullEnumerationContract` class. For for enumeration of all-or-nothing contracts, this solver is mostly applicable for small problem instances ($m < 20$) due to its exponential complexity. In contrast, for threshold contracts the number of possible $\phi$ configurations is linear in $m$, and therefore enumeration is more efficient.

- **Fixed-budget solver**: To find budget-optimal threhsold contracts given a fixed budget $B$, we implement a simple solver which iterates through all possible threshold contracts $t_{j_0}(j) = B \mathbb{1} \left[ j \geq j_0 \right]$ for all $j_0 \in \{0, \ldots, m\}$. The solver then simulates the agent's rational choice (Eq. (2)) and selects the value of $j_0$ which incentivizes the best action. In case of ties between possible values of $j_0$, they are broken in favor of larger values, as this was shown to lead to greater numerical stability. The fixed-budget solver is implemented within the `FixedBudgetThresholdContract` class.

## C.3 Empirical prevalence estimation

Since our theoretical analysis relies on MLRP assumptions, we would like to understand whether the results are applicable to real-world learning curves. Towards this, we run an empirical prevalence evaluation on the MNIST dataset. For each learning algorithm $\mathrm{Alg}$, and training set size $n$, we construct a delegated learning setting with $m = 10$ and collect the following statistics:

- Structural properties:

  - **Is MLRP?** (% MLRP): Check whether all pairs $n_1, n_2 \leq n$ such that $c_{n_1} < c_{n_2}$ satisfy the monotone likelihood ratio property (see Definition 1).

- Computational properties:

  - **SBA successful?** (% SBA): Check whether the SBA algorithm was successful on the given instance.

- Functional form of optimal contract:

  - **Is monotone?** (% M): Check whether the resulting min-budget contract satisfies $t_j \geq t_{j-1}$ for all $j \in [m]$.
  - **Is all or nothing?** (% AoN): Check whether the resulting min-budget contract is an all-or-nothing contract (i.e. whether there exists $j_0 \in \Omega, B > 0$ such that $t_j = 0$ for all $j < j_0$ and $t_j = B$ otherwise).
  - **Is threshold?** (% T): Check whether the resulting min-budget contract is a threshold contract (i.e. whether there exists $j_0 \in \Omega, B > 0$ such that $t_j = 0$ for all $j < j_0$ and $t_j = B$ otherwise).

- Excess cost:

  - **Min-budget**: Optimal objective $B_{\mathrm{LP}}$ of MIN-BUDGET LP without additional restrictions, implemented using the full LP solver.
  - **All-or-nothing budget**: Optimal objective of MIN-BUDGET LP, when the resulting contract is restricted to all-or-nothing form $t_j \in \{0, B_{\mathrm{AoN}}\}$. Implemented using the full enumeration solver.
  - **Threshold budget**: Optimal objective of MIN-BUDGET LP, when the resulting contract is restricted to threshold form $t_j = B_{\mathrm{Thr}} \mathbb{1} \left[ j \geq j_0 \right]$. Implemented using the full enumeration solver.

Table 2: Empirical robustness estimation on the MNIST dataset. Each row corresponds to a learning algorithm, and columns are specified in Appendix C.3. Results are averaged across all implementable actions.

| | Structure | Compute | Functional form of opt. contract | | | Excess cost | |
|---|---|---|---|---|---|---|---|
| | % MLRP | % SBA | % M | % AoN | % T | AoN | T |
| **MLP** | 100% | 94.4% | 100% | 94.4% | 94.4% | 0.04% | 0.04% |
| **GBDT** | 100% | 76.5% | 100% | 82.4% | 82.4% | 0.71% | 0.71% |
| **Logistic** | 93.8% | 81.2% | 93.8% | 93.8% | 93.8% | 0.00% | 0.00% |
| **Perceptron** | 68.8% | 75% | 93.8% | 93.8% | 93.8% | 0.00% | 0.00% |
| **Linear SVM** | 71.4% | 78.6% | 85.7% | 85.7% | 85.7% | 0.95% | 1.45% |
| **Poly SVM** | 100% | 94.4% | 100% | 94.4% | 94.4% | 0.24% | 0.24% |
| **RBF SVM** | 100% | 94.4% | 100% | 94.4% | 94.4% | 0.04% | 0.04% |
| **KNN** | 100% | 100% | 100% | 100% | 100% | 0.00% | 0.00% |
| **Overall** | 92.6% | 87.4% | 97% | 92.6% | 92.6% | 0.22% | 0.26% |

– The excess cost columns in Table 2 indicate the relative excess cost incurred by restricting the min-budget optimization space to simple contracts ($\frac{B_{\{\mathrm{AoN,Thr}\}} - B_{\mathrm{LP}}}{B_{\mathrm{LP}}}$). As all-or-nothing contracts are a superset of threshold contracts, we expect this excess cost to be smaller than the one associated with threshold contracts. The averaging in table 2 is performed on problem instances where both all-or-nothing and threshold contracts are feasible.

Averaging over all implementable actions, we obtain the results in Table 2. The results show that threshold contracts are relatively common in this dataset (more than 85%), and that excess cost of simple contracts is relatively low (around 1%), indicating that threshold contracts may provide good approximation to the optimal min-budget contracts in some cases. All simple optimal contracts in our dataset had a threshold functional form, and some simple optimal contracts were not recovered by SBA, indicating that a tighter characterization of simple contracts may be possible.

Fig. 10 provides qualitative interpretation of the analysis for a selected subset of learners. While the survival functions are not perfectly concave (and thus don't satisfy Concave-MLRP), the SBA algorithm successfully terminates in three cases (MLP, Logistic, KNN). The binding action is $a_{N-1}$ for two of the classifiers (MLP, KNN), similar to the condition implied by Theorem 4. Note that SBA also terminated successfully on Logistic Regression data, despite the learning curve having a distinctive non-concave shape. In the case of GBDT, the survival function is not concave, and the optimal contract does not assume a threshold form.

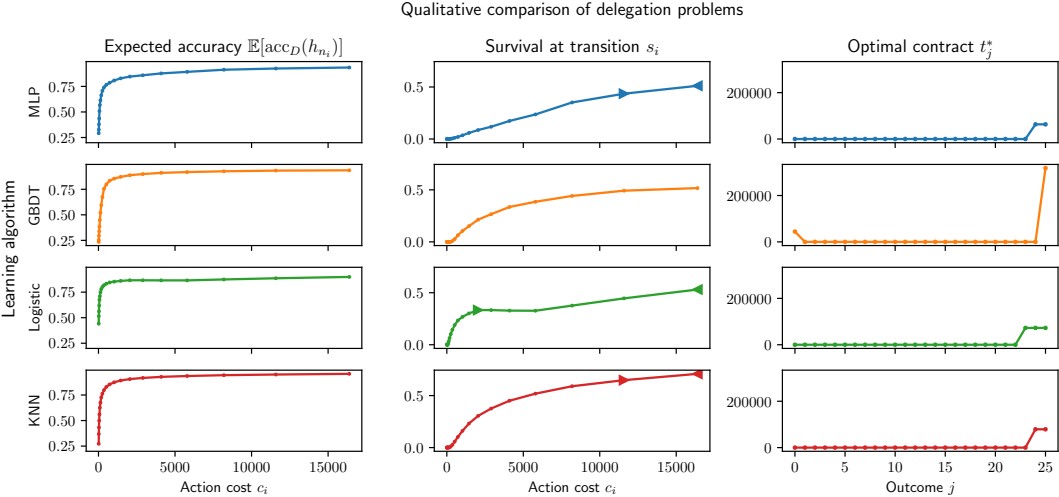

Figure 10: Qualitative comparison of optimal contract computation on LCDB MNIST data ($n^* = 16384$, $m = 25$). Each row represents a learning algorithm. (**Left column**) Expected accuracy curves $\alpha_n$. (**Center column**) Survival function at transition $s_i$, whenever SBA is successful, the binding actions are marked with triangles. (**Right column**) Optimal contract $t_j^*$.

