# OpenReview forum: "Delegated Classification"
_NeurIPS.cc/2023/Conference — NeurIPS 2023 spotlight_

### Official Review · Reviewer_dRCD · 2023-06-22

**Soundness:** 4 excellent
**Presentation:** 4 excellent
**Contribution:** 3 good
**Rating:** 8
**Confidence:** 5

**Summary:**

This work provides a framework for incentive-aware delegation of machine learning tasks. It considers a principal-agent game, where the principal can spend a limited budget on the outsourcing the training of a machine learning model, with the hope of getting the most accurate model, and the agent provides a machine learning model to the principal, but aims to invest minimal effort in training. This work considers the problem of optimal contract design, where the principal commits to a contract that determines how much the agent will be paid for every possible accuracy level of the model, and the agent chooses a profit-maximizing number of samples to use to train the model in response to the contract. This contract design problem requires that the “learning curve”--a function that maps the number of samples to the accuracy of the model is known to both the principal and agent. When the learning curve has a nice structure, it is possible to leverage a connection to the Neyman-Pearson lemma to see that the optimal contracts have a simple threshold form.

**Strengths:**

1. The paper proposes an interesting new problem. The framing of the problem is well-motivated, clear, and sensible.
Figure 2 is a great figure! It nicely captures the motivation for this problem and demonstrates possible contract types.

2. The main contribution of this work is demonstrating that under easy to understand assumptions (Monotone Likelihood Ratio Property, concavity), the optimal contract of the proposed principal-agent game takes on a simple threshold form. The technical contribution of this paper is closely related to the connection between optimal contracts and the Neyman-Pearson lemma established by Bates et al., 2022. It is great to see this work build on the direction of Bates et al, 2022 and demonstrate a connection between optimal contract design and statistical hypothesis testing. The taxonomy given in Table 1 is helpful.

3. The discussion in Lines 352-363 of how overestimation and under-estimation can have quite different implications is useful beyond the scope of this work. It suggests that when estimating learning curves, not all errors should be treated equal and we may want to penalize overestimation more strongly. This may be of interest to communities that aim to estimate/learn scaling laws.


**Weaknesses:**

A potential weakness of the model that the authors propose is the assumption that the principal has access to a learning curve– which captures the stochastic performance of a machine learning model as a function of sample size. Nevertheless, the authors address this weakness quite thoroughly–the authors cite many related works that suggest that the learning curve can be predicted from scaling laws and also analyze the construction of contracts in the partial information setting, where the learning curve is not known and must be estimated.


**Questions:**

1. In Section 2 Lines 110-113, the authors discuss that $h_{n}$ is a stochastic quantity. It would be helpful to add a sentence here that $h_{n}$ is distributed according to a distribution that depends on $n$ (and more details on how to evaluate the expectation will be provided later in the work). At first glance, this equation is somewhat unclear because we have not yet defined the distribution over $h_{n}$ for a given $n$?

2. The second plot of Figure 3 is somewhat hard to parse. Could the authors add some clarifications?

3. It may be helpful to emphasize that the contract $t$ is a function that takes on a different value depending on each possible value of the validation accuracy. (The connection between the domain of $t$ and the sample size of the validation set can easily be missed).

4. What do the authors mean by “robustness” in Line 372?

5. Nit writing: Line 120 “the principal cannot now” -> “the principal cannot know.”

6. Nit: Should the subscript of $f_{n}$ in Equation 2 be $f_{a}$?

7. Nit writing: Line 308 has an incomplete sentence.


**Limitations:**

Yes.

---

> ### Author Rebuttal · Authors · 2023-08-09
>
> Thank you for the encouraging review and positive feedback! We address your questions below:
>
> > In Section 2 Lines 110-113, the authors discuss that $h_n$ is a stochastic quantity. It would be helpful to add a sentence here that $h_n$ is distributed according to a distribution that depends on $n$ (and more details on how to evaluate the expectation will be provided later in the work).
>
> Thank you for this suggestion! We will add a clarification.
>
> > The second plot of Figure 3 is somewhat hard to parse. Could the authors add some clarifications?
>
> Figure 3 (Center) shows the budget required for incentivization of a classifier with accuracy $\\ge 85\\%$, as a function of validation set size $m$. With increasing $m$, the required budget decreases, as the principal is able to gather more information about the agent's chosen action. Asymptotically, the required budget approaches the action cost under full information (“first-best”), indicated by the horizontal dotted lines.
> Solid/dashed curves compare different implementations of the numerical LP solver (local/full solver), as described in Appendix C.2.1.
>
> We believe what may be confusing is that the plot blends economic aspects with optimization aspects. In the final version of the paper we will simplify the plot to focus on the economic aspects, and include a separate discussion on optimization.
>
> > It may be helpful to emphasize that the contract $t$ is a function that takes on a different value depending on each possible value of the validation accuracy. (The connection between the domain of $t$ and the sample size of the validation set can easily be missed).
>
> Thank you for this suggestion! We will clarify this in the paper.
>
> > What do the authors mean by “robustness” in Line 372?
>
> In this context, robustness refers to the ability to design optimal contracts even when the contract design setting has parameters which are uncertain to some extent (e.g when the outcome distribution of each action is estimated from data).
>
> > Nit writing:
> Line 120 “the principal cannot now” -> “the principal cannot know.”;
> Should the subscript of f_n in Equation 2 be  f_a?;
> Line 308 has an incomplete sentence.
>
> Thank you for pointing out these typos! They will be corrected in the forthcoming revision.

---

> > ### Comment · Reviewer_dRCD · 2023-08-13
> >
> > Thanks for these clarifications! I recommend accepting this paper.

---

### Official Review · Reviewer_ybdo · 2023-06-25

**Soundness:** 3 good
**Presentation:** 3 good
**Contribution:** 3 good
**Rating:** 6
**Confidence:** 4

**Summary:**

The paper introduces an interesting problem, and provides interesting theoretical results based on a connection to classical result on statistics. The problem setup follows the standard principal-agent problem with moral hazard, and the principal commits to a contract to incentivize the agent behaving favor of the principal. They mainly analyze that thresholded structure is optimal in case of contract for classification task, and provide some fruitful implications upon it.

**Strengths:**

Overall, the paper is well-written and easy to follow.
The problem setting they considered is novel and of interest to NeurIPS community, especially given the increasing attention to contract theory/delegation mechanism.
The analysis looks sound and the results are thorough.

**Weaknesses:**

I'm not entirely sold on the main results and their implications/takeaways, though I agree its technical soundness.
Details are presented below on questions/limitations.

**Questions:**

Model
* Section 2.1 describes contract design without budget constraint, but the problem setup actually involves budget constraint. I wonder why the authors consider this problem setup, and what happens if the principal does not have a budget constraint (though I understand that both may have plausible applications)
* It seems the agent should be aware of the distribution $f_a$ to compute the expected payment; distribution over possible outcomes from action a. How can this be made practically? Also, $f_a$ corresponds to $f_n(j)$?

Results
* Given the vast literature on contract theory, why the existing techniques cannot be applied to the presented problem setup? I couldn't find the authors discussing on it.

Minor comments
* Reference unresolved in L93

**Limitations:**

* Thresholded rule is optimal - Yes, it makes sense that thresholded contract would be efficient against expected utility maximizer, however, it would carry over a huge amount of variance for the agent's reward. How would the optimal structure change if the agent exhibits risk-averse nature?
* Again, such an extreme thresholded contract is not widely applied in practice - at least some amounts of minimal wage exist, or rather a linear contract is typical (e.g., Carroll15). Besides, finding an agent who admits such a thresholded contract would be difficult than that via posting more conservative/safer contract, thereby possibly inducing the quality of agent to be lower than expected. I'd like to see some discussions on it.
* In the above context, if the budget constraint is given in ex-post manner, given the use of constant/linear/threshold with the same maximal payment (as in Fig 2), I doubt that the quality of the agent would be endogenous depending on the pricing rule to be exploited (e.g., giving constant B would attract highest-quality agent), rather than being exogenous. I'd appreciate some (empirical or theoretical) discussions on it.
* As the paper contributes to an applied modeling of real-world scenario with claiming the efficiency of thresholded rule, I expected to see some comparisons between various contract structures in the experiment.
* Also, the title of delegated classification looks a bit overly abstract to me. I would (though weakly) suggest making it more explicit, e.g., including contract sort of notions.

---

> ### Author Rebuttal · Authors · 2023-08-09
>
> Thank you for the positive feedback and insightful questions!
>
> > What happens if the principal does not have a budget constraint?
>
> In the non-budgeted setting, and when MLRP holds, min-pay optimal contracts take on a rather extreme form: a single non-zero payment that can be arbitrarily large and arbitrarily rare [DRT18]. In particular, Appendix B.3 shows a concrete setting where the maximal payment of an optimal non-budgeted contract is exponential in the validation set size $m$. Every entity has bounds on the payment it can make, and exponential growth makes such contracts unscalable. Given that MLRP is likely to hold for reasonable learning tasks, we believe this justifies the exploration of budgeted contracts.
>
> Without a budget constraint - and if willing to forgo global optimality - a linear contract may be a viable alternative, as such contracts are known to have appealing robustness properties (e.g. [C15]). However, they also have significant shortcomings, such as arbitrarily-inferior performance compared to optimal contracts [DRT18]. They also require the principal to attach a precise monetary value to each possible outcome, which for some settings may be unrealistic (e.g., when the principal is a government or non-profit organization).
>
> A contract design that requires the principal only to set a budget, and guarantees the best-in-class classifier given this budget, is thus arguably more natural in many situations. We will add to the paper a discussion comparing the different approaches.
>
> > It seems the agent should be aware of the distribution $f_a$ to compute the expected payment; distribution over possible outcomes from action a. How can this be made practically?
>
> We envision two avenues through which the distribution $f_a$ could be estimated by the agent. First, prior experience - as a service provider, the agent is likely to have past experience training classifiers in different scenarios, and thus may have the ability to generalize across learning curves. This approach has been applied in the empirical learning curves literature [LB05]. Second, extrapolation - similar to the partial information model introduced in Sec. 4.2, the agent may also train exploratory models and make a decision based on the extrapolated learning curve. Due to the agent's ability to collect data and train models in-house, we expect their extrapolation quality to be higher than that of the principal.
>
> > Also, $f_a$ corresponds to $f_n(j)$?
>
> Thank you, there was a typo in eq. (2), and $f_a$ should come in place of $f_n$.
>
> $f_n(j)$ is the $j$-th component in the outcome probability distribution corresponding to action $n$.
>
> > Given the vast literature on contract theory, why the existing techniques cannot be applied to the presented problem setup?
>
> Existing results from the contract design literature cannot be directly applied because min-budget contract design problems have a different structure. We discuss similarities and differences in Appendix B.3, and will expand the discussion in the forthcoming revision. Our LP duality proof is distinct but inspired by [DRT18], and understanding whether additional techniques can be transferred from the theory of budget-unconstrained contracts to min-budget ones is a very interesting direction for future work.
>
> > Reference unresolved in L93
>
> Fixed. Thanks for pointing this out!
>
> > How would the optimal structure change if the agent exhibits risk-averse nature?
>
> This is an interesting question, for which the answer is not immediate. For comparison, note that in auction theory, answering questions regarding the incorporation of behavioral aspects has warranted substantial empirical and theoretical work, e.g. [VW21]. Our work focuses on a standard rational agent model, which we believe is reasonable as a first step. Hopefully our work can help establish the necessary foundations for extending to more elaborate agent models.
>
> > Such an extreme thresholded contract is not widely applied in practice - at least some amounts of minimal wage exist, or rather a linear contract is typical (e.g., Carroll15).
>
> Thanks for this question! While working on it, we found a proof showing that the rationality assumption is made without loss of generality with respect to minimum wage: We show that any min-budget contract with minimum wage can be represented as a sum of a min-budget contract (eq. (4)), and a constant representing the minimum wage. In other words, an optimal contract with minimum wage can be obtained by adding a constant to a solution of eq. (3). The proof follows by constructing a modified version of the MIN-BUDGET linear program (eq. (5)), and introducing a change of variables. The full statement and proof will be added to the paper.
>
> We also add that threshold contracts are common in many practical scenarios. For example, a salesperson may get a bonus upon selling a certain amount of units, and a student may earn a certificate upon exceeding a certain grade average.
>
> > Besides, finding an agent who admits such a thresholded contract would be difficult than that via posting more conservative/safer contract, thereby possibly inducing the quality of agent to be lower than expected.
>
> Your question seems to imply a more elaborate setup in which there exists a population of agents, and where agents have types (i.e., “quality”). Again, this is a very intriguing question, but remains outside our scope.
>
> > I expected to see some comparisons between various contract structures in the experiment.
>
> In Appendix B.3, we compare the performance of min-budget to min-pay contracts, illustrating the impracticality of the latter as discussed above.
>
> —
>
> References:
> * [LB05] Leite & Brazdil, 2005 - Predicting relative performance of classifiers from samples
> * [C15] Caroll, 2015 - Robustness and Linear Contracts
> * [DRT18] Dütting et al., 2018 - Simple versus Optimal Contracts
> * [VW21] Vasserman & Watt, 2021 - Risk Aversion and Auction Design: Theoretical and Empirical Evidence

---

> > ### Comment · Reviewer_ybdo · 2023-08-16
> >
> > Thanks for your response.
> > I have no further questions at the moment.

---

### Official Review · Reviewer_au5P · 2023-07-07

**Soundness:** 4 excellent
**Presentation:** 4 excellent
**Contribution:** 3 good
**Rating:** 5
**Confidence:** 4

**Summary:**

This paper presents a novel theoretical principal-agent framework for examining the incentive-aware delegation of machine learning tasks. In this context, a principal can design a monetary contract to stimulate an agent to exert private efforts towards training a classifier. In the proposed framework, the agent's private actions is the number of samples he/she can process, while the principal operates within a monetary budget $B$. Importantly, the contract design is entirely based on observed outcomes, modeled as the prediction accuracy of the agent's trained classifier against the validation samples.

The paper first shows that the budget-optimal contract is essentially an all-or-nothing contract when the agent has only binary actions. Then the paper also provides several (structural) characterizations on the optimal contract if the action-to-outcome distributions satisfy certain regularity assumptions (e.g., monotone likelihood ratio property, concavity). Finally, the paper also provides empirical results to study how the model parameters affect the design of the contracts and the resulting agent’s trained classifier.

**Strengths:**

The paper is rigorous and very well-written. I largely view this work as an modeling paper. The proposed principal-agent framework is novel and interesting, and it provides an economic view to understand the dynamics when a training task is delegated to another entity, potentially possessing conflicting interests. The paper also provides several characterizations on the optimality of contract when the considered problem has some certain structure (e.g., binary-action, binary-outcome), and also some characterizations for the general action/outcome space but with imposing some additional assumptions. The empirical section also adequately evaluates the proposed framework.

**Weaknesses:**

My main concern regarding the paper lies within the technique results. It is noted that many of the characterizations about the optimal contract align closely with, or can be derived from, recent research on the algorithmic principal-agent problem.

For instance, as noted by the author (line 262), the optimal contract for binary-action can be derived using the Linear Programming (LP) duality. The paper instead uses a proof approach related to hypothesis testing. It could enhance the paper's value if the author elucidated the advantages of this particular approach. Can it illuminate more complex instances, and if so, how?

In addition, in my humble opinion, it seems that the current characterizations could also be derived by merely considering a pure principal-agent problem, thereby bypassing any classification or machine learning elements. Since the primary aim of this paper is to frame the delegated training problem as a principal-agent problem, I believe the exploration could be enriched if it delves deeper into how typical tradeoffs (e.g., the number of samples used by the agent has some implications on the prediction accuracy in a quantitive way via some generalization error) in standard machine learning tasks affect the considered game.

Other questions:

1. missing refs in Line 93
2. it is a bit confusing in Proposition 2, is that $B$ the budget? If so, then by definition of all-or-nothing contract, there should be only one outcome that has positive payment?
3. In Program (5) in Appendix B.2, the objective should be $t$?

**Questions:**

See above

---

> ### Author Rebuttal · Authors · 2023-08-09
>
> Thank you for the detailed review and insightful feedback!
>
> > My main concern regarding the paper lies within the technique results. It is noted that many of the characterizations about the optimal contract align closely with, or can be derived from, recent research on the algorithmic principal-agent problem.
>
> This is imprecise, and we apologize if this was the impression conveyed. Let us clarify:
>
> The contract design literature deals predominantly with minimizing expected payment. This turns out to be inappropriate for modeling delegation of learning (see response to reviewer ybdo), and motivates the introduction of *budget-optimal contracts*. To the best of our knowledge, there is no prior work on budget-optimal contracts, and therefore existing results do not necessarily apply.
>
> Budgets make the contract design problem challenging because they introduce an additional hard constraint. At the same time, properties of the underlying learning task, which motivate monotonicity and concavity, can make the problem feasible (see below). The fact that budget-optimal contracts admit a simple form (“all-or-nothing”) is a novel result, which we prove using duality theory (see below). Such techniques are also used in min-pay contracts, but the connection is not immediate.
>
> > For instance, as noted by the author (line 262), the optimal contract for binary-action can be derived using the Linear Programming (LP) duality. The paper instead uses a proof approach related to hypothesis testing.
>
> This is also imprecise, though in re-reading this section, we see how this could have been implied from our formulation of line 262 (which we will correct). We apologize, and again wish to clarify:
>
> The connection to hypothesis testing is not a proof technique, but rather a result:
> * We prove Prop. 2 (optimal contract for binary-action) by a stand-alone proof that relies entirely on LP duality (Appx. B.4.1).
> * Prop. 2 is not directly connected to hypothesis testing, but the functional form of the optimal contract suggests a possible connection to Neyman-Pearson (NP).
> * Based on this observation, we formalize the equivalence between optimal contracts and optimal tests in Thm. 1, where one direction of the proof makes use of Prop. 2 (Appx. B.4.2).
>
> “Loose” connections between moral hazard and hypothesis testing have been known for some time, but formal evidence has been limited: (i) the maximal likelihood ratio appearing in optimal min-pay contracts [S15], and (ii) a connection established by Bates et al. [BJSS22] in a setting distinct from ours (adverse selection, rather than moral hazard), and to a different variant of the NP lemma (asymmetric rather than symmetric). Thm. 1 provides a direct correspondence between optimal contracts and optimal hypothesis tests, and as such, establishes a clean formal connection. Our hope is that this will enable tighter future connections, and transfer between the two domains.
>
> We will clarify both of the above points in the forthcoming revision of the paper.
>
> > It seems that the current characterizations could also be derived by merely considering a pure principal-agent problem, thereby bypassing any classification or machine learning elements.
>
> The structure of a delegated contract derives from the structure of the underlying machine learning problem, coupled with the incentives of the different parties involved. For example:
> * Expected outcomes in the delegated task are determined by the *learning curve* of the underlying learning task. The study of learning curves has drawn recent interest in the learning community (e.g., see [VL22]). Our results leverage recent results in this field, such as scaling laws [K20] and monotonization [BDKMMS22].
> * The variance in outcomes in delegated learning is induced (in part) by the randomness of the training set $S$. This is also a key element in PAC analysis, manifested differently in our setup (i.e., as variance rather than as a probabilistic guarantee).
> * The principal evaluates outcomes using a *validation set*. This affects the design of the contract by imposing structure that we leverage. Validation is a fundamental concept in learning, and to the best of our knowledge, is unique to delegation of ML.
>
> All of the above play important roles in our modeling choices and analysis.
>
> > I believe the exploration could be enriched if it delves deeper into how typical tradeoffs … in standard machine learning tasks affect the considered game.
>
> We fully agree. In Sec. 3, our analytic results rely on several basic properties of learning tasks, as discussed above. In Sec. 4, we move beyond this, and empirically study how additional elements of the underlying classification problem affect delegation and its outcomes, such as:
> * Using "empirical" contracts, computed on the basis of an outcome-probability matrix estimated from finite data
> * Extrapolating to large-sample actions from small-sample data, and for different extrapolation approaches
> * Differences in contracts and outcomes for the same data but using different model classes (e.g., MLP vs. GBDT)
> * Sensitivity and implications of different error types (i.e., under- vs. over-estimation) for the principal
>
> > Missing refs in Line 93
>
> Fixed. Thank you!
>
> > In Proposition 2, is $B$ the budget?
>
> Yes. In all-or-nothing contracts, payment takes one of two values ($t_j\\in\\{0,B\\}$), and multiple $t_j$’s can be positive.
>
> > In Program (5), the objective should be $t$?
>
> Objective is $B$ because we minimize the $l^\infty$ norm of $t$. This is similar to a technique presented in [BV14; Sec. 4.3.1].
>
> —
>
> References:
> * [VL22] Viering & Loog, 2022 - The shape of learning curves: a review
> * [K20] Kaplan et al., 2020 - Scaling Laws for Neural Language Models
> * [BDKMMS22] Bousquet et al., 2022 - Monotone Learning
> * [S15] Salanié, 2015 - The Economics of Contracts: A Primer (Chapter 5)
> * [BJSS22] Bates et al., 2022 - Principal-Agent Hypothesis Testing
> * [BV14] Boyd & Vandenberghe, 2004 - Convex Optimization

---

> > ### Comment · Reviewer_au5P · 2023-08-19
> >
> > Thank the author for responding my questions. I do not have further questions.

---

### Official Review · Reviewer_EHyc · 2023-07-24

**Soundness:** 3 good
**Presentation:** 4 excellent
**Contribution:** 3 good
**Rating:** 7
**Confidence:** 3

**Summary:**

The paper studies the problem of a decision maker (the principal) delegating the task of training a machine learning model to an agent.  Both parties are strategic, and the principal must commit to a contract to encourage the agent to invest effort (e.g., labeling training samples).  The authors consider the principal's problem of designing the optimal contract subject to a budget constraint, which maximizes the expected performance of the model trained by the agent given that the agent best responds to the contract.  Technically, they give (1) an LP-based algorithm for computing the optimal contract, (2) a characterization connecting the optimal contract to hypothesis testing when the agent has only 2 possible actions, and (3) another characterization stating that optimal contracts have a simple form when the learning curve satisfies certain conditions.  They validate these findings with experiments, and also empirically study a setting where the principal does not have enough information and therefore must estimate the learning curve.

**Strengths:**

The problem is natural and exhibits quite some theoretical depth.  The paper is well written, and in particular, the introduction nicely motivates the problem.  The technical results are nice and clean, and they also appear to be practically meaningful, as supported by experimental results.

**Weaknesses:**

While I like the paper overall, one concern is regarding the model: the model is quite specific, and I wonder if / to what extent the results can be generalized and remain (approximately) valid.  Another practical concern is regarding scalability: in the experimental section the authors say that their full LP solver works for m no larger than 20, which doesn't sound like a very practical number; there is the local solver but it's not totally clear what it does, or how reliable it is.  Also see detailed comments for some minor points.

**Questions:**

(also putting minor comments here)

Figure 1: at this point it's not totally clear to me what role m plays in the model.  In particular, how should m be chosen (either by nature or by the principal), and how does the choice affect the performance of the contract?  I'm sure this will become clear later, but it might make sense to briefly comment earlier, perhaps in a footnote.

Line 56, "MLRP": what does this mean?

Line 93: broken citation

Line 120: "... principal cannot now how many examples ..."

Line 130: "a-priory"

Line 266, "under MLRP n_2 is always implementable": I'm not sure I get this --- what if c_2 - c_1 > B?

Line 272, "important special case of binary-outcome": this (corresponding to m = 1) sounds less practical to me.  Any justification for the importance of this case?

Theorem 2: I feel the way this result is presented in Table 1 is somewhat misleading.  The impression I initially got from the table is that the problem of computing the unconditional, unrestricted optimal contract is NP-hard.  Or is this actually implied by Theorem 2?

Line 307, "as m increases, required budgets": unfinished sentence?

Line 310, "the local solver is easy to run": what does the local solver do?

---

> ### Author Rebuttal · Authors · 2023-08-09
>
> Thank you for the feedback and the great suggestions! We address your questions below:
>
> > Figure 1: at this point it's not totally clear to me what role $m$ plays in the model. In particular, how should $m$ be chosen (either by nature or by the principal), and how does the choice affect the performance of the contract? I'm sure this will become clear later, but it might make sense to briefly comment earlier, perhaps in a footnote.
>
> Thanks for the suggestion. We will add clarification for the role of $m$ in the caption of figure 1, and in the problem setup section.
>
> > Line 56, "MLRP": what does this mean?
>
> MLRP stands for “monotone likelihood ratio property”. It is a common assumption in statistics [LR05] and contract design literature [DRT18]. In our setting, the MLRP assumption can be taken to state (informally) that the better the evaluation result of the classifier, the more likely that it was trained using more data. A formal definition of MLRP is given in Line 264. We will clarify the definition and the intuition behind it, and better connect the two.
>
> > Line 93: broken citation;
> Line 120: "... principal cannot now how many examples ...";
> Line 130: "a-priory"
>
> Typos fixed. Thank you for pointing them out!
>
> > Line 266, "under MLRP n_2 is always implementable": I'm not sure I get this --- what if c_2 - c_1 > B?
>
> The meaning of “implementable” is that the min-budget optimization problem (eq. (4)) is guaranteed to have a solution. The min-budget optimization problem finds the minimal budget $B^*$ required for incentivizing training with a certain dataset size (in the case described above - $n_2$). We also note that $B^* \\ge c_2 - c_1$ for every implementable contract incentivizing $n_2$, as the expected payout must cover the agent’s additional cost.
>
> $B$ represents the total budget available to the principal, and we treat it as an external constraint in order to maintain parity with the contract design literature (e.g [S15]). When $B^*>B$, the solution is considered economically infeasible.
>
> > Line 272, "important special case of binary-outcome": this (corresponding to m = 1) sounds less practical to me. Any justification for the importance of this case?
>
> This case captures a coarse measurement of the agent’s performance - the agent’s effort can either succeed or fail. It is an important case since it is a primary focus of many works in the literature, such as [BFNW12, HSV16, DEFK21]. It is also practical in situations where the principal can either accept the agent’s work and pay the agreed upon price, or altogether reject the work and not pay.
>
> > Theorem 2: I feel the way this result is presented in Table 1 is somewhat misleading. The impression I initially got from the table is that the problem of computing the unconditional, unrestricted optimal contract is NP-hard. Or is this actually implied by Theorem 2?
>
> Table 1 focuses on characterization of contracts that take a simple form. In our context, simple-form contracts have the “all-or-nothing” property ($t_j \\in \\{0, B\\}$), as opposed to paying arbitrary amounts ($t_j \\in [0,B]$). Theorem 2 implies that optimal simple contracts are NP-hard to find in the general case, by reduction from 3-SAT. In contrast, the problem of computing the unconditional, unrestricted optimal contract is not NP-hard. We will make sure to clarify this better in the next version.
>
> > Line 307, "as m increases, required budgets": unfinished sentence?
>
> Thanks for pointing this out! This was a typo (now fixed), and the complete sentence is “As $m$ increases, required budgets and the difference between them both decrease.”.
>
> > Line 310, "the local solver is easy to run": what does the local solver do?
>
> The local solver computes the optimal contract using a relaxed version of the MIN-BUDGET linear program (eq. (5)).
>
> In more detail, the MIN-BUDGET linear program (eq. (5)) can in principle be solved with general-purpose numerical solvers (e.g. GLPK), but is prone to numerical instabilities; we observed these starting at around $m\approx 20$. To overcome this, the local solver relaxes some of the incentive compatibility constraints (Appendix C.2.1). This leads to a closed-form solution (given by Proposition 2) which can be computed efficiently and at scale. Under the conditions of Theorem 3, namely MLRP + concave survival, the local solver is provably optimal. We will clarify this and add a discussion of the different solvers in the revision.
>
> —
>
> References:
> * [LR05] Lehman & Romano, 2005 - Testing Statistical Hypotheses (Section 3.4)
> * [DRT18] Dütting et al., 2018 - Simple versus Optimal Contracts
> * [S15] Salanié, 2015 - The economics of contracts: a primer (Chapter 5)
> * [DEFK21] Dütting et al., 2021 - Combinatorial Contracts
> * [HSV16] Ho et al., 2016 - Adaptive Contract Design for Crowdsourcing Markets
> * [BFNW12] Babaioff et al., 2012 - Combinatorial Agency

---

> > ### Comment · Reviewer_EHyc · 2023-08-18
> >
> > Thank you for your response!  I have no further questions.

---

### Decision · Program_Chairs · 2023-09-21

**Decision:**

Accept (spotlight)

**Comment:**

The reviewers were in agreement regarding this paper. The paper studies a model of incentive-aware design for delegated classification, which may be of interest to folks in the federated learning community. Some of the paper's strong traits are: the modeling approach, the rigorous theoretical approach, and the presentation. The paper also evaluates the results empirically, although there was some reviewer-author discussion about the scalability of the empirical results. The authors agreed to add a discussion about this, so I don't think it's going to be a problem for the final version. Their key contribution is theoretical. I'm suggesting this paper for a spotlight; although the techniques are more Econ-oriented, I think the conference audience will greatly benefit from being exposed to the model and the idea of using contract theory to solve it.